# RGMDT: Return-Gap-Minimizing Decision Tree Extraction in Non-Euclidean Metric Space

**Jingdi Chen**
The George Washington University
jingdic@gwu.edu

**Hanhan Zhou**
The George Washington University
hanhan@gwu.edu

**Yongsheng Mei**
The George Washington University
ysmei@gwu.edu

**Carlee Joe-Wong**
Carnegie Mellon University
cjoewong@andrew.cmu.edu

**Gina Adam**
The George Washington University
ginaadam@gwu.edu

**Nathaniel D. Bastian**
United States Military Academy
nathaniel.bastian@westpoint.edu

**Tian Lan**
The George Washington University
tlan@gwu.edu

## Abstract

Deep Reinforcement Learning (DRL) algorithms have achieved great success in solving many challenging tasks while their black-box nature hinders interpretability and real-world applicability, making it difficult for human experts to interpret and understand DRL policies. Existing works on interpretable reinforcement learning have shown promise in extracting decision tree (DT) based policies from DRL policies with most focus on the single-agent settings while prior attempts to introduce DT policies in multi-agent scenarios mainly focus on heuristic designs which do not provide any quantitative guarantees on the expected return. In this paper, we establish an upper bound on the return gap between the oracle expert policy and an optimal decision tree policy. This enables us to recast the DT extraction problem into a novel non-euclidean clustering problem over the local observation and action values space of each agent, with action values as cluster labels and the upper bound on the return gap as clustering loss. Both the algorithm and the upper bound are extended to multi-agent decentralized DT extractions by an iteratively-grow-DT procedure guided by an action-value function conditioned on the current DTs of other agents. Further, we propose the Return-Gap-Minimization Decision Tree (RGMDT) algorithm, which is a surprisingly simple design and is integrated with reinforcement learning through the utilization of a novel Regularized Information Maximization loss. Evaluations on tasks like D4RL show that RGMDT significantly outperforms heuristic DT-based baselines and can achieve nearly optimal returns under given DT complexity constraints (e.g., maximum number of DT nodes).

38th Conference on Neural Information Processing Systems (NeurIPS 2024).

# 1 Introduction

Deep Reinforcement Learning (DRL) has significantly advanced real-world applications in various domains [5, 12, 13, 16, 17, 29, 44, 69, 71]. However, the black-box nature of DRL's deep neural networks, with their multi-layered structures and millions of parameters, makes them largely uninterpretable. This lack of interpretability is particularly problematic in safety-critical sectors such as healthcare and aviation, where clarity in machine decision-making is crucial [14, 15, 27, 31, 43, 45, 51].

Decision trees (DTs) address this problem by supporting rules and decision lists that enhance human understanding [1, 11, 35, 36, 38, 56]. However, there are two main challenges to applying DTs in DRL: (1) DTs extracted from DRL often lack performance guarantees [2, 22]. There have been imitation learning-type approaches that train DTs with samples drawn from the trained DRL policy [3], but the return gap is characterized in terms of number of samples needed and does not apply to DTs with arbitrary size constraints (e.g., on the maximum number of nodes $L$). (2) Although learning DT policies for interpretability has been investigated in the single-agent RL setting [37, 42, 55, 58], it is under-explored in the multi-agent setting, except for centralized methods like MAVIPER [48], **which lacks a performance guarantee**. In multi-agent, decentralized settings where rewards are jointly determined by all agents' local DTs, any changes in one agent's DT will impact the optimality of other agents' DTs. The return gap for such decentralized DTs has not been considered.

In this paper, we propose a DT extraction framework, a Return-Gap-Minimization Decision Tree (RGMDT) algorithm, that is proven to achieve a closed-form guarantee on the expected return gap between a given RL policy and the resulting DT policy. **Our key idea** is that each decision path of the extracted DT maps a subset of observations to an action attached to the leaf node. Thus, constructing a DT can be considered an iterative clustering problem of the observations into different decision paths, with actions at leaf nodes as labels. Since the action-value function $Q(o, a)$ represents the potential future return, it can be leveraged to obtain a return gap bound for the process. We show that it recasts the DT extraction problem as a non-Euclidean clustering with respect to a loss defined using $Q(o, a)$. Due to its iterative structure, RGMDT supports an iterative algorithm to generate return-gap-minimizing DTs of arbitrary size constraints.

Further, we extend our algorithm to **multi-agent settings** and provide a performance bound on the return gap. A naïve approach that simply converts each agent's (decentralized) policy into a local DT cannot ensure global optimality of the resulting return since the reward is jointly determined by the DTs constructed for all agents. We develop an **iteratively-grow-DT** process, which iteratively identifies the best step to grow the DT of each agent, conditioned on the current DTs of other agents (**by revising the resulting action-value function**) until the desired complexity is reached. Thus, the impact of decentralized DTs on the joint reward is captured during this process. The method ensures that we learn decentralized DTs yet also guarantees a return gap.

More precisely, we show that the problem is recast into a **non-Euclidean clustering** problem in the observation space with $n$ agents. Each agent $j$ computes an updated action-value function by conditioning it on other agents' current DTs (thus capturing the impact of their DT constructions on each other). Then, guided by this updated action-value-function, agent $i$ grows its DT by adding another label (i.e., decision path and corresponding leaf node) by a clustering function $\mathcal{T} = [\mathcal{T}_i(a_i|o_i, \mathcal{T}_{-i}), \forall i]$, which is conditioned on its local observations $o_i$ and the DT policies of other agents $\mathcal{T}_{-i}$. The process continues iteratively across all agents until their decentralized DTs reach the desired size constraints. To analyze the resulting return gap, we show that the difference between the given DRL policy and the DTs can be characterized using a Policy Change Lemma, relying on the average distance between joint action values in each cluster (i.e., decision path). Intuitively, if the observations following the same decision path and arriving at the same leaf node are likely maximized with the same action, the DT policy would have a small return gap with the DRL policy $\pi^*$. We show that the return gap is bounded by $O(\sqrt{\epsilon/(\log_2(L+1) - 1)}nQ_{\max})$, with respect to the maximum number of leaf nodes $L$ of the DT, the number of agents $n$, the highest action-value $Q_{\max}$ and the average cosine-distance $\epsilon$ between joint action-value vectors corresponding to the same clusters. The result allows us to minimize this upper bound to find the optimal DTs.

The main **contributions** are: (1). We quantify the return gap between an oracle DRL policy and its extracted DTs via an upper bound with respect to the DT size constraints. (2). We propose RGMDT, an algorithm that constructs multi-agent decentralized DTs of arbitrary sizes by minimizing the upper

bound of the return gap. Instead of drawing samples from the DRL policy for DT learning, RGMDT recasts the problem as an iterative non-Euclidean clustering problem of the observations into different decision paths, with actions at leaf nodes as labels. (3). Rather than generating a DT and subsequently applying pruning algorithms to achieve the desired complexity, the RGMDT framework constructs DTs of any given size constraint while minimizing the upper bound of the resulting expected return gap. (4). We show that RGMDT significantly outperforms baseline DT algorithms and can be applied to complex tasks like D4RL with significant improvement.

The remainder of this paper is organized as follows. Section 2 reviews related work on decision trees and their integration with reinforcement learning, with a focus on interpretability and multi-agent systems. Section 3 introduces our problem formulation and defines the return gap. In Section 4, we present the theoretical results, recasting return gap minimization as clustering in a non-Euclidean space, and deriving bounds for both single-agent and multi-agent settings. Section 5 details the construction of SVM-based decision trees and the iteratively growing framework for minimizing return gaps, with pseudocode in the appendix. Section 6 describes the experimental setup and presents empirical results. Finally, Section 7 concludes with a summary of contributions and future directions.

## 2 Related Work

**Effort on Interpretability for Understanding Decisions.** To enhance interpretability in decision-making models, one strategy involves crafting interpretable reward functions within inverse reinforcement learning (IRL), as suggested by [10, 67, 68]. This approach offers insights into the underlying objectives guiding the agents' decisions. Agent behavior has been conceptualized as showing preferences for certain counterfactual outcomes [7], or as valuing information differently when under time constraints [34]. However, extracting policies through black-box reinforcement learning (RL) algorithms often conceals the influence of observations on the selection of actions. An alternative is to directly define the agent's policy function with an interpretable framework. Reinforcement learning policies have thus been articulated using a high-level programming language [60], or by framing explanations around desired outcomes [66], facilitating a more transparent understanding of decision-making processes.

**Interpretable RL via Decision Tree-based models.** Since their introduction in the 1960s, DTs have been crucial for interpretable supervised learning [47, 50, 53]. The CART algorithm [8], established in 1984, is foundational in DT methodologies and underpins Random Forests (RF) [9] and Gradient Boosting (GB) [23], which are benchmarks in predictive modeling. These techniques are central to platforms like ranger and scikit-learn and continue to evolve, as seen in the iterative random forest, which explores stable interactions in data [4, 52, 64, 70]. To interpret an RL agent, Frosst et al [25] explain the decisions made by DRL policies by using a trained neural network to create soft decision trees. Coppens et al. [19] propose distilling the RL policy into a differentiable DT by imitating a pre-trained policy. Similarly, Liu et al. [39] apply an imitation learning framework to the Q-value function of the RL agent. They also introduce Linear Model U-trees (LMUTs), which incorporate linear models in the leaf nodes. Silva et al. [57] suggest using differentiable DTs directly as function approximators for either the Q function or the policy in RL. Their approach includes a discretization process and a rule list tree structure to simplify the DTs and enhance interpretability. Additionally, Bastani et al. [3] propose the VIPER method, which distills policies as neural networks into a DT policy with theoretically verifiable capabilities that follow the Dataset Aggregation (DAGGER) method [54], specifically for imitation learning settings and nonparametric DTs. Ding et al. [20] try to solve the instability problems when using imitation learning with tree-based model generation and apply representation learning on the decision paths to improve the decision tree-based explainable RL results, which could achieve better performance than soft DTs. Milani et al. extend VIPER methods into multi-agent scenarios [48] in both centralized and decentralized ways, they also summarize a paper about the most recent works in the fields of explainable AI (artificial intelligence) [49], which confirms the statements that small DTs are considered naturally interpretable.

However, traditional DT methods are challenging to integrate with RL due to their focus on correlations within training data rather than accounting for the sequential and long-term implications in dynamic environments. Imitation learning approaches have been explored in single-agent RL settings [42, 55, 58]. For instance, VIPER [3] trains DTs with samples from a trained DRL policy, but its return gap depends on the number of samples needed and does not apply to DTs with arbitrary size constraints (e.g., maximum number of nodes $L$). DT algorithms remain under-explored in multi-agent

settings, except for MAVIPER [48] that extends VIPER to multi-agent scenarios[46] while lacking performance guarantees and remaining a heuristic design.

# 3 Preliminaries and Problem Formulation

A **Dec-POMDP** [6] models cooperative MARL, where agents lack complete information about the environment and only have local observations. We formulate a Dec-POMDP as a tuple $D = \langle S, A, P, \Omega, n, R, \gamma \rangle$, where $S$ is the joint **state** space and $A = A_1 \times A_2 \times \cdots \times A_n$ is the joint **action** space, where $\mathbf{a} = (a_1, a_2, \ldots, a_n) \in A$ denotes the joint action of all agents. $P(\mathbf{s}'|\mathbf{s}, \mathbf{a}) : S \times A \times S \to [0, 1]$ is the **state transition function**. $\Omega$ is the **observation** space. $n$ is the total number of agents. $R(\mathbf{s}, \mathbf{a}) : S \times A \to \mathbb{R}$ is the **reward function** in terms of state $\mathbf{s}$ and joint action $\mathbf{a}$, and $\gamma$ is the discount factor. Given a policy $\pi$, we consider the average expected return $J(\pi) = \lim_{T \to \infty} (1/T) E_\pi[\sum_{t=0}^{T} R_t]$. The goal of this paper is to minimize the return gap between the pre-trained RL policy providing action-values $\pi^* = [\pi_i^*(a_i|o_1, \ldots, o_n), \forall i]$ and the DT policy $\mathcal{T} = [\mathcal{T}_i(a_i|o_i, \mathcal{T}_{-i}), \forall i]$ where decision tree policies $\mathcal{T}_{-i} = \{\mathcal{T}_j = \phi_j(o_j, l_j), \forall j \neq i\}$, where $l_j = g(o_j)$, and the function $g$ is the clustering function for observation $o_j$. Define the **return gap** as:

$$\min J(\pi^*) - J(\mathcal{T}). \tag{1}$$

While the problem is equivalent to maximizing $J(\mathcal{T})$, the return gap can be analyzed more easily by contrasting $\mathcal{T}$ and $\pi^*$. We derive an upper bound of the return gap and then design efficient clustering strategies to minimize it. We consider the discounted observation-based state value and the corresponding action-value functions for the Dec-POMDP:

$$V^\pi(\mathbf{o}) = \mathbb{E}_\pi\Big[\sum_{i=0}^{\infty} \gamma^i \cdot R_{t+i}\Big|\mathbf{o}_t = \mathbf{o}, \mathbf{a}_t \sim \pi\Big], Q^\pi(\mathbf{o}, \mathbf{a}) = \mathbb{E}_\pi\Big[\sum_{i=0}^{\infty} \gamma^i \cdot R_{t+i}\Big|\mathbf{o}_t = \mathbf{o}, \mathbf{a}_t = \mathbf{a}\Big], \tag{2}$$

where $t$ is the current time step. Re-writing the average expected return as an expectation of $V^\pi(\mathbf{o})$:

$$J(\pi) = \lim_{\gamma \to 1} E_\mu[(1 - \gamma)V^\pi(\mathbf{o})], \tag{3}$$

where $\mu$ is the initial observation distribution at time step $t = 0$, i.e., $\mathbf{o}(0) \sim \mu$. We will leverage this state-value function $V^\pi(\mathbf{o})$ and its corresponding action-value function $Q^\pi(\mathbf{o}, \mathbf{a})$ to unroll the Dec-POMDP and derive a closed-form upper-bound to quantify the return gap.

# 4 Theoretical Results and Methodology

We first recast the problem as clustering in a **non-Euclidean Metric Space** and then prove a single-agent result. Then we expand the single-agent result to multi-agent settings. Since directly minimizing the return gap is intractable, we bound the performance of RGMDT with the return gap between the *oracle* **policy** $\pi^* = [\pi_i^*(a_i|o_1, \ldots, o_n), \forall i]$ **corresponding to obtaining the action-values** $Q^{\pi^*}$ and optimal decision tree policy $\mathcal{T}^L = [\mathcal{T}_1^L, \ldots, \mathcal{T}_n^L]$, where each $\mathcal{T}_i$ **can only have** $L$ **nodes**, and $n$ is the total number of agents. For simplicity, we use $V^*$ to represent $V^{\pi^*}$, and $Q^*$ to represent $Q^{\pi^*}$. We assume that observation/action spaces defined in the Dec-POMDP tuple are discrete with finite observations and actions, i.e., $|\Omega| < \infty$ and $|A| < \infty$. For Dec-POMDPs with continuous observation and action spaces, the results can be easily extended by considering cosine-distance between action-value functions and replacing summations with integrals, or sampling the action-value functions as an approximation.

**Lemma 4.1.** *(Policy Change Lemma.) For any policies $\pi^*$ and DT policy $\mathcal{T}^L$ with $L$ leaf nodes, the optimal expected average return gap is bounded by:*

$$J(\pi^*) - J(\mathcal{T}^L) \leq \sum_l \sum_{\mathbf{o} \sim l} [Q^*(\mathbf{o}, \mathbf{a}_t^{\pi^*}) - Q^{\mathcal{T}^L}(\mathbf{o}, \mathbf{a}_t^{\mathcal{T}^L})]d_\mu^{\mathcal{T}^L}(\mathbf{o}),$$
$$d_\mu^{\mathcal{T}^L}(\mathbf{o}) = (1 - \gamma) \sum_{t=0}^{\infty} \gamma^t \cdot P(\mathbf{o}_t = \mathbf{o}|\mathcal{T}^L, \mu), \tag{4}$$

*where $d_\mu^{\mathcal{T}^L}(\mathbf{o})$ is the $\gamma$-discounted visitation probability under decision tree $\mathcal{T}^L$ and initial observation distribution $\mu$, and $\sum_{\mathbf{o} \sim l}$ is a sum over all observations corresponding to the decision path from the parent node to the leaf node $l$, where $l$ indicates its class.*

**Proof Sketch.** Our key idea is to leverage the state value function $V^{\mathcal{T}^L}(\mathbf{o})$ and its corresponding action-value function $Q^{\mathcal{T}^L}(\mathbf{o}, \mathbf{a})$ in Eq.(2) to unroll the Dec-POMDP from timestep $t = 0$ and onward. Detailed proof is provided in the Appendix.

Then we define the **action-value vector** corresponding to observation $o_j$, i.e.,

$$\bar{Q}^*(o_j) = [\widetilde{Q}^*(o_j, \mathbf{o}_{-j}), \forall o_{-j}], \tag{5}$$

where $\mathbf{o}_{-j}$ are the observations of all other agents and $\widetilde{Q}^*(o_j, \mathbf{o}_{-j})$ is a vector of action-values weighted by marginalized visitation probabilities $d_\mu^\pi(\mathbf{o}_{-j}|o_j)$ and corresponding to different actions, i.e., $\widetilde{Q}^*(o_j, \mathbf{o}_{-j}) = [Q^*(o_j, \mathbf{o}_{-j}, \pi^*(a_j|o_j), \mathbf{a}_{-j}) \cdot d_\mu^\pi(\mathbf{o}_{-j}|o_j)]$. **At the initial iteration step of the iteratively-grow-DT process**, since we did not grow the DT for agent $j$ yet, we use *oracle* policy $\pi^*$ to give a deterministic action $a_j^* = \text{argmax}_{a_j} Q_j^*(o_j, a_j)$ based on $o_j$ for obtaining action-value vectors, and $\mathbf{a}_{-j}$ are all possible actions for agents except for agent $j$, which makes $\widetilde{Q}^*(o_j, \mathbf{o}_{-j})$ a vector where each entry corresponds to the full set of actions $\mathbf{a}_{-j}$ across all agents except for agent $j$.

Next, constructing decision trees can be considered an iterative clustering problem of the observations into different decision paths $l_j = g(o_j)$, with actions at leaf nodes as labels. Then, the $o_j$ are divided into clusters, each labeled with the clustering label $l_j$ to be used for constructing DT. We bound the policy gap between $\pi_{(j)}^*$ and $\mathcal{T}_{(j)}^L$, using the average cosine-distance of action-value vectors $\bar{Q}^*(o_j)$ corresponding to $o_j$ in the same cluster and its cluster center $\bar{H}(l) = \sum_{o_j \sim l} \bar{d}_l(o_j) \cdot \bar{Q}^*(o_j)$ under each label $l$. Here $\bar{d}_l(o_j) = d_\mu^{\mathcal{T}^L}(o_j)/d_\mu^{\mathcal{T}^L}(l)$ is the marginalized probability of $o_j$ in cluster $l$ and $d_\mu^{\mathcal{T}^L}(l)$ is the probability of label $l$ under DT $\mathcal{T}^L$, and the environments' initial observation distribution is represented by $\mathbf{o}(t = 0) \sim \mu$. To this end, we let $\epsilon(o_j) = D_{cos}(\bar{Q}^*(o_j), \bar{H}(l))$ be the cosine-distance between vectors $\bar{Q}^*(o_j)$ and $\bar{H}(l)$ and consider the **average cosine-distance** $\epsilon$ across all clusters represented by different clustering labels $l$ for one iteration of growing DT:

$$\epsilon \triangleq \sum_l d_\mu^\pi(l) \sum_{o_j \sim l} \bar{d}_l(o_j) \cdot \epsilon(o_j), \tag{6}$$

The result is summarized in Thm. 4.2.

**Theorem 4.2.** *(Impact of Decision Tree Conversion.) Consider two optimal policies $\pi_{(j)}^*$ and $\mathcal{T}_{(j)}^L$ obtained from the policy providing action-values and the DT, the optimal expected average return gap is bounded by:*

$$J(\pi_{(j)}^*) - J(\mathcal{T}_{(j)}^L) \leq O(\sqrt{\epsilon/(\log_2(L+1) - 1)} Q_{\max}) \tag{7}$$

*where $Q_{\max}$ is the maximum absolute action-value of $\bar{Q}^*(o_j)$ in each cluster as $Q_{\max} = max_{o_j} ||\bar{Q}^*(o_j)||_2$, and $\epsilon$ is the average cosine-distance defined in Eq.(6), $L$ is maximum number of leaf nodes of the resulting DT.*

**Proof Sketch.** We give an outline below and provide the proof in the Appendix.

*Step 1: Recasting DT construction into a Non-Euclidean Clustering.* Viewing the problem as a clustering of $o_j$, restrict policy $\mathcal{T}_{(j)}$ (conditioned on $l_j$) to take the same actions for all $o_j$ in the same cluster under the same label $l_j$. We perform clustering on the observation $o_j$ within the observation space $\Omega_j$ by grouping the corresponding action-value vectors $\bar{Q}^*(o_j)$ in the action-value vector space $\mathcal{Q}_j$, using the cosine-distance function $D_{\cos}$ as the clustering metric. We demonstrate that $(\Omega_j, D_{\cos})$ constitutes a **Non-Euclidean Metric Space**. The findings are detailed in Lemma 4.3, with the full proof provided in the Appendix.

**Lemma 4.3.** *(Cosine Distance Metric Space Lemma.) Let $\Omega_j$ be a set of observations with associated vector representations in $\mathbb{R}^m$ obtained through a mapping function $\bar{Q}^*$, and let $D_{cos} : \Omega_j \times \Omega_j \to \mathbb{R}$ be a distance function defined as:*

$$D_{cos}(o_j^a, o_j^b) = 1 - f(\bar{Q}^*(o_j^a), \bar{Q}^*(o_j^b)) = 1 - \frac{\bar{Q}^*(o_j^a) \cdot \bar{Q}^*(o_j^b)}{\bar{Q}^*(o_j^a)|| \cdot ||\bar{Q}^*(o_j^b)||} \tag{8}$$

*where $f$ denotes the cosine similarity. Then, the pair $(\Omega_j, D_{cos})$ forms a metric space. The proof is in Appendix.*

*Step 2: Rewrite the return gap in vector form.* Re-writing the optimal expected average return gap derived in Policy Change Lemma 4.1 in vector terms using action-value vectors $\bar{Q}^*(o_j)$ and an auxiliary maximization function $\Phi_{\max}(\bar{Q}^*(o_j))$ that returns the largest component of vector $\bar{Q}^*(o_j)$:

$$J(\pi^*_{(j)}) - J(\mathcal{T}_{(j)}) \leq \sum_l d^\pi_\mu(l)[\sum_{o_j \sim l} \bar{d}_l(o_j) \cdot \Phi_{\max}(\bar{Q}^*(o_j)) - \Phi_{\max}(\sum_{o_j \sim l} \bar{d}_l(o_j) \cdot \bar{Q}^*(o_j))], \quad (9)$$

*Step 3: Projecting action-value vectors toward cluster centers.* By projecting $\bar{Q}^*(o_j)$ toward $\bar{H}(l)$, $\bar{Q}^*(o_j)$ could be re-written as $\bar{Q}^*(o_j) = Q^\perp(o_j) + \cos\theta_{o_j} \cdot \bar{H}_l$, then we could upper bound $\Phi_{max}(\bar{Q}^*(o_j))$ by:

$$\Phi_{\max}(\bar{Q}^*(o_j)) \leq \Phi_{\max}(\cos\theta_{o_j} \cdot \bar{H}_l) + \Phi_{\max}(Q^\perp(o_j)).$$

Taking a sum over all $o_j$ in the cluster, we have $\sum_{o_j \sim l} \bar{d}_l(o_j)\Phi_{\max}(\cos\theta_{o_j} \cdot \bar{H}_l) = \Phi_{\max}(\bar{H}_l)$, since the projected components $\cos\theta_{o_j} \cdot \bar{H}_l$ should add up to exactly $\bar{H}_l$. To bound Eq.(9)'s return gap, it remains to bound the orthogonal components $Q^\perp(o_j)$.

*Step 4: Deriving the upper bound w.r.t. cosine-distance.* We derive an upper bound on the return gap by bounding the orthogonal projection errors using the average cosine distance within each cluster.

$$\Phi_{max}(Q^\perp(o_j)) \leq O(\sqrt{\epsilon(o_j)}Q_{\max}). \quad (10)$$

Using the concavity of the square root with Eq.(6), we derive the desired upper bound $J(\pi^*_{(j)}) - J(\mathcal{T}_{(j)}) \leq O(\sqrt{\epsilon}Q_{\max})$ **for one iteration of growing the DT**, then **for $I = \log_2(L+1) - 1$ iterations** ($I$ is the depth of the DT), the upper bound is $O(\sqrt{\epsilon/(\log_2(L+1)-1)}Q_{\max})$. Here we assume that the DT is a perfectly balanced or full binary tree. If it is a complete binary tree, the number of iterations is $I = \log_2(L)$. In the worst-case scenario of a highly unbalanced tree, the number of iterations is $I = L - 1$. Then we extend this upper bound to multi-agent settings in Thm. 4.4.

**Theorem 4.4.** *In $n$-agent Dec-POMDP, the return gap between policy $\pi^*$ corresponding to the obtained action-values and decision tree policy $\mathcal{T}^L$ conditioned on clustering labels is bounded by:*

$$J(\pi^*) - J(\mathcal{T}^L) \leq O(\sqrt{\epsilon/(\log_2(L+1)-1)}nQ_{\max}). \quad (11)$$

**Proof Sketch.** Beginning from $\pi^* = [\pi^*_i(a_i|o_1, o_2, \ldots, o_n), \forall i]$, we can construct a sequence of $n$ policies, each replacing the conditioning on $o_j$ by constructed decision tree $\mathcal{T}^L_j$, for $j = 1$ to $j = n$, one at a time. This will result in the decision tree policies $\mathcal{T}^L = [\mathcal{T}^L_j(a_j|o_j, \mathcal{T}^L_{-j})]$. Applying Thm. 4.2 for $n$ times, we prove the upper bound between $J(\pi^*)$ and $J(\mathcal{T}^L)$ for multi-agent scenarios.

*Remark* 4.5. Thm. 4.4 holds for any arbitrary finite number of leaf nodes $L$. Furthermore, increasing $L$ reduces the average cosine distance (since more clusters are formed) and, consequently, a reduction in the return gap due to the upper bound derived in Thm. 4.4.

## 5   Constructing SVM-based Decision Tree to Minimize Return Gaps

The result in Thm. 4.4 inspires a **iteratively-grow-DT** framework - RGMDT, which constructs return-gap-minimizing multi-agent decentralized DTs of arbitrary sizes. This is because RGMDT grows a binary tree iteratively (in both single and multi-agent cases) until the desired complexity is reached. The method addressed **two challenges**: (1). RGMDT constructs the optimal DT that minimizes the return gap given the complexity of the DT (e.g., the number of the leaf nodes). (2). RGMDT addressed the scalability problems of multi-agent DT construction with provided theoretical guarantees. We summarize the pseudo-code in the Appendix.

**Non-Euclidean Clustering Labels Generation.** We approximate the non-Euclidean clustering labels $l_j = g(o_j)$ for each agent using DNNs parameterized by $\xi = \{\xi_1, \ldots, \xi_n\}$. Prior to growing the decision tree (DT) for each agent $j$, we sample a minibatch of $K_1$ transitions $\mathcal{X}_j$ from the replay buffer $\mathcal{R}$, which includes observation-action-reward pairs from all agents. We then identify the top $K_2$ most frequent observations $\mathbf{o}^{k_2}_{-j}$ in $\mathcal{X}_j$, and retrieve their associated actions using the pre-trained policy $\pi^*$, forming a set $\mathcal{X}_{-j}$. We combine these samples with $(o_j, \pi^*(o_j))$ from $\mathcal{X}_j$ to form the

dataset $\mathcal{D}$ for training. The *oracle* critic networks, parameterized by $\omega$, compute the action-values for this dataset, approximating the vectors $\bar{Q}^*(o_j)$. $g_{\xi_j}(o_j)$ is updated by optimizing a Regularized Information Maximization (RIM) loss function [32]:

$$\mathcal{L}(g_{\xi_i}) = \sum_{p=1}^{K_1} \sum_{q \in N_{K_3}(p)} \left[ D_{cos}(\bar{Q}^*(o_j^p), \bar{Q}^*(o_j^q)) \, \|l_j^p - l_j^q\|^2 - [H(m_j) - H(m_j|o_j)] \right], \quad (12)$$

the first term is a locality-preserving clustering loss, which enhances the cohesion of clusters by encouraging action-value vectors close to each other to be grouped together. This is achieved using the cosine distance $D_{\text{cos}}$ to identify the $K_3$ nearest neighbors of each action-value vector. The second term, the mutual information loss, quantifies the mutual information between the observation $o_j$ and the cluster label $m_j$. It aims to balance cluster size and clarity by evaluating the difference between the marginal entropy $H(m_j)$ and the conditional entropy $H(m_j|o_j)$.

**Single-agent DT Construction via Iterative Clustering.** To grow a decision tree (DT) for agent $j$, each decision path of RGMDT maps a subset of observations to an action attached to the leaf node (**decision paths for different leaf node counts are visualized in Appendix**). The process iteratively groups observations and assigns actions as labels at each path's end. The main challenge is determining the optimal division of observations at each node and defining the split conditions. **For each split**, the RGMDT framework uses a sample matrix of observations $\mathbf{X} \in \mathbb{R}^{d \times n}$ and associated class labels $\mathbf{Y} \in \mathbb{R}^{1 \times n}$, where $n$ and $d$ are the number of examples and dimensions, respectively. Each label $l_i = g(o_i)$ corresponds to a category $\{l_1, l_2, \ldots, l_L\}$, which is generated from **non-euclidean clustering function** $g$ with **cosine-distance loss function** $\mathcal{L}(g_{\xi_i})$, and $L$ being the maximum number of leaf nodes. We use **Support Vector Machines (SVM)** [41] to identify the optimal hyperplane $\mathcal{H}(\mathbf{X})$ that maximizes the margin between the closest class points, known as support vectors. This results in the optimized hyperplane $\mathcal{H}^*(\mathbf{X})$ defined by $\mathbf{w}^* \cdot \mathbf{o} - p^* = 0$. The DT, structured as a binary tree for multi-class classification, incorporates this SVM-derived hyperplane. Each node splits into two child nodes based on the hyperplane's criteria: left for $\mathbf{w}^* \cdot \mathbf{o} - p^* < 0$ and right for $\mathbf{w}^* \cdot \mathbf{o} - p^* \geq 0$. **Note that the splits are still based on linear thresholds.** The construction of the $\mathcal{T}_j$ is complete until it reaches the maximum number of leaf nodes $L$ by **iteratively repeating the splitting and clustering for $I = \log_2(L) + 1$ iterations**. The child node in the last iterations will become a leaf node assigned the class label $l = \arg\max_i \left\{ \frac{n_i}{r} \mid i = l_1, l_2, \ldots, l_L \right\}$, where $r$ is the total number of sample points in the current leaf node, and $n_i$ is the number of sample points in class $i$. Classification can be conducted based on the $\mathcal{T}_j$. For an observation with an unknown class, it constantly goes through the split nodes and finally reaches a leaf node where $l$ indicates its class [62].

**Iteratively-Growing-DT for Multi-agent.** In each iteration, we obtain a revised $Q(\mathbf{o}, \mathbf{a})$ by conditioning it on the current DTs of all other agents, i.e., if two actions are merged into the same decision path, then the $Q(\mathbf{o}, \mathbf{a})$ are merged too. Then we grow the DT as guided by the revised $Q(\mathbf{o}, \mathbf{a})$, which iteratively identifies the best step to grow the DT of each agent conditioned on the current DTs of other agents. To simplify, consider agents $i \neq j$ as a conceptual super agent $-j$. With a constraint of $L$ leaf nodes, we iteratively grow DTs for agent $j$ and super agent $-j$, growing one more level for each agent per iteration with a total of $\log_2(L) + 1$ iterations. At the initial iteration ($i = 0$), for agent $j$, we calculate $\widetilde{Q}^*_{i=0}(o_j, \mathbf{o}_{-j}) = [Q^*(o_j, \mathbf{o}_{-j}, \pi^*(a_j|o_j), \mathbf{a}_{-j}) \cdot d_\mu^\pi(\mathbf{o}_{-j}|o_j)]$ and grow DT $\mathcal{T}_j^{i=0}$ using the clustering function $g(o_j)$. For super agent $-j$, the DT is based on $\widetilde{Q}^*_{i=0}(\mathbf{o}_{-j}, o_j)$ which integrates the DT from agent $j$. Each subsequent iteration updates the DTs by recalculating $\widetilde{Q}^*$ using the prior iteration's DTs to ensure consistent action choices within clusters for both agents.

# 6 Evaluation and Results

**Experiment Setup and Baselines.** We test RGMDT on both **discrete** and **continuous** state space problems in the maze environments and the **D4RL** [26]. To explore how well RGMDT scales in **multi-agent** environments, we also constructed similar target-chasing tasks in the maze following the same settings as the Predator-Prey tasks in the Multi-Agent Particle Environment (MPE) [40] (detailed in Appendix). We use Centralized Q Learning and Soft Actor-critic (SAC) [30] to obtain the action-value vectors. We compare RGMDT against strong **baselines**. The baselines include **different types of Imitation DTs** [48]: Each DT policy is directly trained using a dataset collected by running the expert policies for multiple episodes. No resampling is performed. The observations of an agent are the features, and the actions of that agent are the labels. **(1). Single Tree Model**

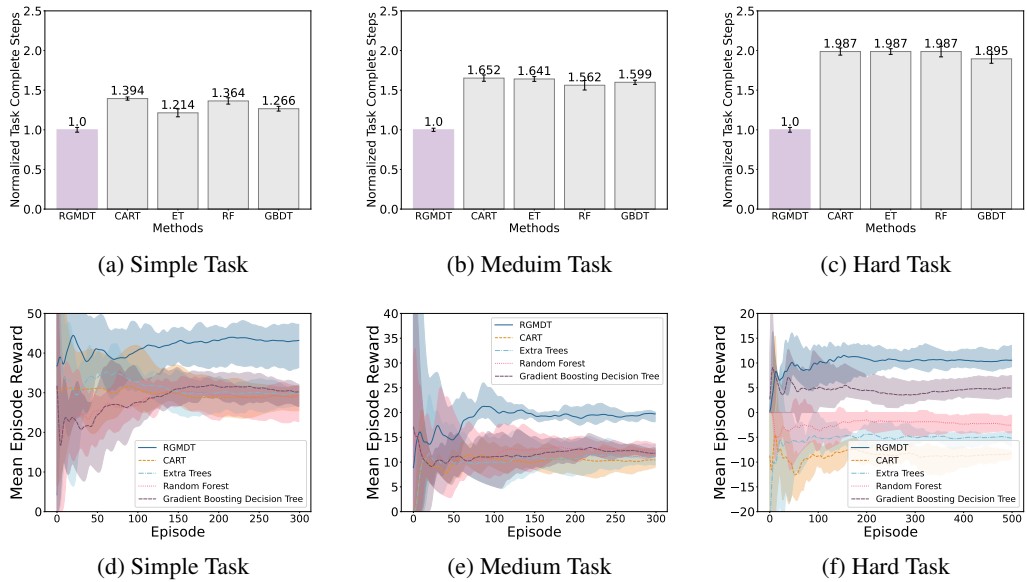

(a) Simple Task       (b) Meduim Task       (c) Hard Task

(d) Simple Task       (e) Medium Task       (f) Hard Task

Figure 1: Evaluation on Maze tasks. (a)-(c): RGMDT (purple bar) completes the tasks in fewer steps than all the baselines. (d)-(f): RGMDT (blue line) achieves a higher mean episode reward than all the baselines in all scenarios with varying complexities, which illustrates its ability to minimize the return gap in hard environments.

using **CART** [8], which is the most famous traditional DT algorithm whose splits are determined by choosing the best cut that maximizes the gain. **(2).** DT algorithms with Bootstrap Aggregating **error correction methods**: **Random Forest (RF)** [9] that reduces variance by averaging multiple deep DTs, trained on different parts of the same training set; **Extra Trees (ET)** [28] that uses the whole dataset to grow the trees which make the boundaries more randomized. **(3). Boosting For Error Correction**, **Gradient Boosting DTs** [24] that builds shallow trees in succession where each new tree helps to correct errors made by the previously trained tree. **(4). Multi-agent Verifiable Reinforcement Learning via Policy Extraction (MAVIPER):** A centralized DT training algorithm that jointly grows the trees of each agent [48], which extends VIPER [3] in multi-agent settings. All the experiments are **repeated 3-5 times** with different seeds. More details about the evaluations are in the Appendix.

**Single Agent Task.** In the single-agent task, an agent is trained to navigate to a target without colliding with walls to complete the task. We increase the complexity of the environment and evaluate the RGMDT across three different levels of complexity (detailed in Appendix). For each experiment, we compare agents' performance using RGMDT against baselines using two metrics: the number of time steps required to complete the task (fewer is better) and the mean episode rewards (higher is better). We conducted each experiment five times for both metrics using different random seeds to ensure robustness.

Figures 1a-1c illustrate the **normalized task completion steps** for all methods relative to RGMDT's average performance. RGMDT consistently outperforms the baselines by completing tasks in fewer steps, increasing its advantage in more complex tasks. In the most challenging task, where only 10 steps are allowed in complete the hard task, RGMDT typically finishes in about 5 steps,

Table 1: RGMDT achieved higher rewards with a smaller sample size and node counts on D4RL.

| Algorithm | 80,000 samples | | 800,000 samples | |
|---|---|---|---|---|
| | 40 nodes | 64 nodes | 40 nodes | 64 nodes |
| **RGMDT** | $\mathbf{567.27 \pm 160.47}$ | $\mathbf{776.70 \pm 150.18}$ | $\mathbf{467.27 \pm 134.78}$ | $\mathbf{458.70 \pm 98.07}$ |
| CART | $443.47 \pm 153.49$ | $448.85 \pm 154.12$ | $458.47 \pm 131.48$ | $460.90 \pm 90.40$ |
| RF | $345.81 \pm 178.43$ | $452.89 \pm 134.76$ | $456.41 \pm 124.23$ | $489.92 \pm 71.69$ |
| ET | $196.55 \pm 147.92$ | $448.98 \pm 119.40$ | $441.01 \pm 138.43$ | $451.80 \pm 83.24$ |

whereas the baselines often fail, resulting in normalized steps that are twice as high as RGMDT. Figures 1d to 1f display the **mean episode reward** curves over 300/500 episodes for methods tested post-offline training with DTs of maximum 4 nodes in 3 types of tasks. RGMDT consistently outperforms all baselines, with the performance improvement becoming more obvious as task complexity increases. In the simplest task, both RGMDT and baseline DTs earn positive rewards. However, RGMDT shows a significant performance advantage in the medium complexity task. In the most challenging task, while most baselines struggle to complete the task, RGMDT completes it and achieves higher rewards. The results show RGMDT's effectiveness in minimizing the negative effects of fewer decision paths, thereby maintaining its performance in increasingly complex environments.

Figure 2 demonstrates the impact of leaf node counts on RGMDT's performance in the hard task, comparing mean episode rewards for RGMDT and baselines with $|L| = 4, 8, 16, 32$ leaf nodes. RGMDT's performance improves with an increasing number of leaf nodes. Notably, with just $|L| = 4$ leaf nodes, RGMDT is the only method to complete the task and outperform all baselines. With more than $|L| = 4$ leaf nodes, RGMDT's performance approaches the near-optimal levels of the expert RL policy. This performance supports Thm. 4.4's prediction that return gaps are bounded by $O(\sqrt{(\log_2(L) + 1)\epsilon}nQ_{\max})$, and that these gaps decrease as more leaf nodes reduce the average cosine-distance across clusters, as noted in Remark 4.5. **Decision paths for different leaf node counts are visualized in the Appendix.**

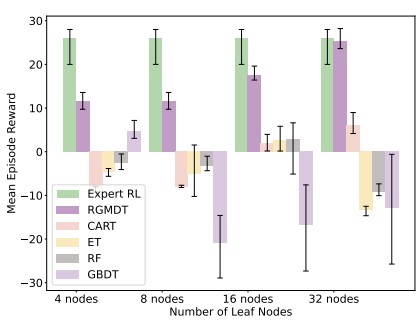

Figure 2: The normalized mean episode reward increases as the total number of leaf nodes increases(Hard Maze).

**D4RL.** Table. 1 presents the achieved rewards for four algorithms (RGMDT, CART, RF, and ET) on the Hooper problem instance from the D4RL datasets [26], evaluated across different training sample sizes (800,000 and 80,000) and DT node counts (40 and 64). Each cell shows the achieved reward and its standard deviation. The bold values indicate that RGMDT achieves higher rewards than all baselines, especially with smaller sample sizes and node counts. When trained with $800,000$ samples, RGMDT performs better when the node counts are reduced from $64$ to $40$, while the baselines' achieved rewards decrease. When the sample size is reduced tenfold to $80,000$, RGMDT achieves $69.33\%$ and $21.40\%$ improvements in the achieved rewards with $64$ and $40$ nodes, respectively, whereas other baselines suffer from the sample size reduction. It shows that RGMDT provides a succinct, discrete representation of the optimal action-value structure, leading to less noisy decision

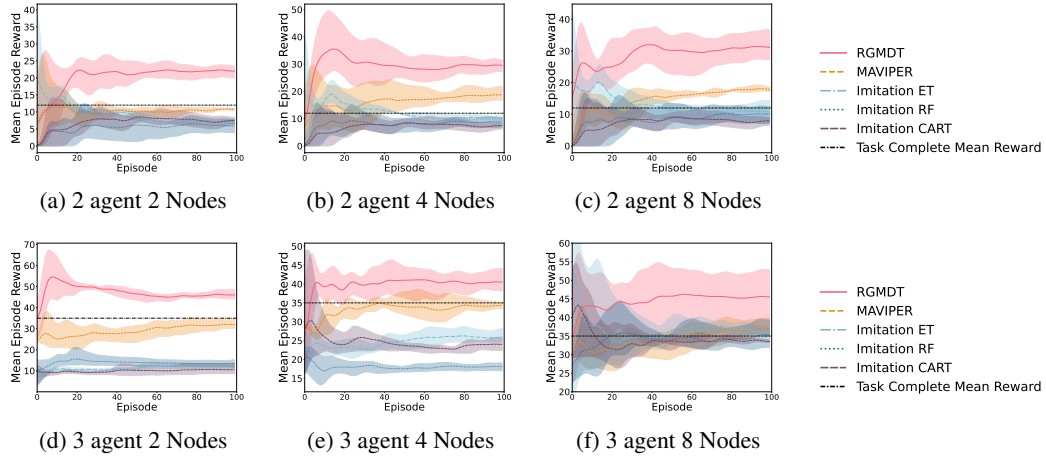

Figure 3: Comparisons on the $n$-agent tasks: (a)-(c) 2 agents, (d)-(f) 3 agents. RGMDT with limited leaf nodes can learn these tasks much faster than the baselines and have better final performance, even when most of the baselines fail on the task with 2 and 4 leaf nodes.

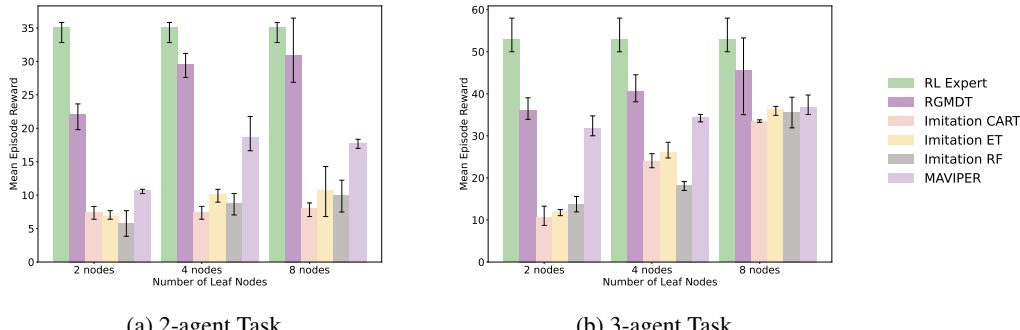

| (a) 2-agent Task | (b) 3-agent Task |

Figure 4: The reward of RGMDT (starred purple bar) outperforms all baselines and increases with the number of leaf nodes, achieving performance comparable to the expert RL (no hatch style bar).

Table 2: All changes contributed to the improvement of RGMDT, especially the non-Euclidean clustering and iteratively-grow-DT designs.

| Description | Task Completed | Mean Reward |
| --- | --- | --- |
| Described RGMDT model | **Yes** | $\mathbf{41.95 \pm 3.08}$ |
| Removed SVM hyperplane, using CART | No | $26.16 \pm 3.74$ |
| Removed SVM hyperplane, using ET | Yes | $36.79 \pm 2.15$ |
| Removed SVM hyperplane, using RF | No | $21.59 \pm 3.81$ |
| Removed SVM hyperplane, using GBDT | Yes | $37.76 \pm 2.42$ |
| Removed Non-Euclidean-Clustering Module | **No** | $23.85 \pm 2.23$ |
| Removed iteratively-grow-DT process | **No** | $16.29 \pm 4.72$ |

paths and allowing agents to discover more efficient decision-making DT conditioned on the non-euclidean clustering labels with smaller sample sizes and fewer nodes.

**Multi-Agent Task.** In Figures 3a-3c, RGMDT outperforms all baselines, particularly as the number of leaf nodes decreases. Notably, with just 2 leaf nodes, all baselines fail to complete the task, whereas RGMDT succeeds with significantly higher rewards. This performance trend continues in more challenging scenarios, as shown in Figures 3d-3f, where no baseline can complete the 3-agent task with 4 leaf nodes. This demonstrates RGMDT's capability to adapt to fewer leaf nodes and more complex environments by efficiently utilizing information from action-value vectors. Figure 4 confirms that RGMDT's performance improves with more leaf nodes, achieving near-optimal levels with $|L| = 4$ or more, consistent with the findings of Thm. 4.4 and supporting its application in multi-agent settings as noted in Remark 4.5.

**Ablation Study:** Table. 2 compares various configurations of the RGMDT model in a multi-agent setting, highlighting task completion and mean rewards. Notably, RGMDT completed tasks with an average reward of $41.95 \pm 3.08$. Configurations without using SVM to derive the linear splitting hyperplane at each split and employing algorithms like CART, ET, RF, and GBDT show varied success: ET and GBDT complete tasks, while CART and RF cannot. Removing the Non-Euclidean Clustering Module or the iteratively-grow-DT specifically designed for multi-agent contexts results in task failures, particularly the latter, which records the lowest mean reward. Results show all changes improved RGMDT, especially the non-Euclidean clustering and iteratively-grow-DT designs.

## 7   Conclusion and Future Works

This paper introduces an iteratively-grow-DT framework for MARL, which views clustering label generation as a non-euclidean clustering problem and quantifies the optimal return gap between the given RL policy and the resulting DT policy with a closed-form upper bound. We propose a novel class of DT algorithm, RGMDT, designed to minimize the return gap using limited leaf nodes. RGMDT significantly outperforms the baselines and achieves nearly optimal returns. Further research will delve into the regression tree that is more suitable for the continuous state/action spaces, and we will also include the re-sampling module to enhance our algorithm.

## Acknowledgements

This work was supported in part by the U.S. Military Academy (USMA) under Cooperative Agreement No. W911NF-22-2-0089, Office of Naval Research (ONR) grants N00014-23-1-2850 and N00014-24-1-2073. The views and conclusions expressed in this paper are those of the authors and do not reflect the official policy or position of the U.S. Military Academy, U.S. Army, U.S. Department of Defense, U.S. Government, or the perspectives of the Office of Naval Research.

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

## A   Impact Statements

The proposed framework can iteratively grow decision trees for multi-agent environments, while we do not see an immediate threat of ethical and societal consequences, it is essential to address and mitigate any risks of misuse or unintended consequences from applying malicious training datasets or improper use of certain scenarios.

## B   Additional Related Work Discussion

**Effort on Interpretability for Understanding Decisions.** To enhance interpretability in decision-making models, one strategy involves crafting interpretable reward functions within inverse reinforcement learning (IRL), as suggested by [10]. This approach offers insights into the underlying objectives guiding the agents' decisions. Agent behavior has been conceptualized as showing preferences for certain counterfactual outcomes [7], or as valuing information differently when under time constraints [34]. However, extracting policies through black-box reinforcement learning (RL) algorithms often conceals the influence of observations on the selection of actions. An alternative is to directly define the agent's policy function with an interpretable framework. Reinforcement learning policies have thus been articulated using a high-level programming language [60], or by framing explanations around desired outcomes [66], facilitating a more transparent understanding of decision-making processes.

**Decision Trees.** Since their introduction in the 1960s, DTs have been crucial for interpretable supervised learning [47, 50, 53]. The CART algorithm [8], established in 1984, is foundational in DT methodologies and underpins Random Forests (RF) [9] and Gradient Boosting (GB) [23], which are benchmarks in predictive modeling. These techniques are central to platforms like ranger and scikit-learn and continue to evolve, as seen in the iterative random forest, which explores stable interactions in data [4, 52, 64, 70]. However, traditional DT methods are challenging to integrate with RL due to their focus on correlations within training data rather than accounting for the sequential and long-term implications in dynamic environments. Imitation learning approaches have been explored in single-agent RL settings [42, 55, 58]. For instance, VIPER [3] trains DTs with samples from a trained DRL policy, but its return gap depends on the number of samples needed and does not apply to DTs with arbitrary size constraints (e.g., maximum number of nodes $L$). DT algorithms remain under-explored in multi-agent settings, except for MAVIPER [48] that extends VIPER to multi-agent scenarios[46] while lacking performance guarantees and remaining a heuristic design.

**Interpretable RL via Tree-based models.** To interpret an RL agent, Frosst et al [25] explain the decisions made by DRL policies by using a trained neural net to create soft decision trees. Coppens et al. [19] propose distilling the RL policy into a differentiable DT by imitating a pre-trained policy. Similarly, Liu et al. [39] apply an imitation learning framework to the Q-value function of the RL agent. They also introduce Linear Model U-trees (LMUTs), which incorporate linear models in the leaf nodes. Silva et al. [57] suggest using differentiable DTs directly as function approximators for either the Q function or the policy in RL. Their approach includes a discretization process and a rule list tree structure to simplify the trees and enhance interpretability. Additionally, Bastani et al. [3] propose the VIPER method, which distills policies as neural networks into a DT policy with theoretically verifiable capabilities that follow the Dataset Aggregation (DAGGER) method proposed in [54], specifically for imitation learning settings and nonparametric DTs. Ding et al. [20] try to solve the instability problems when using imitation learning with tree-based model generation and apply representation learning on the decision paths to improve the decision tree-based explainable RL results, which could achieve better performance than soft DTs. Milani et al. extend VIPER methods into multi-agent scenarios [48] in both centralized and decentralized ways, they also summarize a paper about the most recent works in the fields of explainable AI [49], which confirms the statements that small DTs are considered naturally interpretable.

## C   Background and Preliminaries

### C.1   Decision Tree

**Multivariate Decision Tree:** Wang et al. [61] introduced two variants of multivariate decision tree classifiers structured as full binary trees: the Randomly Partitioned Multivariate Decision

Tree (MDT1) and the Principal Component Analysis (PCA)-Partitioned Multivariate Decision Tree (MDT2). MDT1 and MDT2 utilize a unified approach to construct a binary tree through top-down recursive partitioning, ceasing when no further nodes require division. Initially, the root node encompasses all training data. The process simplifies as (1). Split the current node into left and right children using a multivariate hyperplane. (2). Apply a split condition; nodes meeting this condition are further partitioned, while others become leaf nodes, adopting the majority class for prediction. (3). Proceed to the next node for partitioning, repeating from step 1 until no dividable nodes remain.

The framework focuses on two main aspects: constructing the multivariate hyperplane for each node's split and defining the split condition. We'll detail these aspects by examining a single split, involving a node with a sample matrix $X \in \mathbb{R}^{d \times n}$ and corresponding class labels $Y \in \mathbb{R}^{1 \times n}$, where $n$ and $d$ represent the number of examples and dimensions, respectively. Each label $y_i \in \{l_1, l_2, ..., l_c\}, i = 1, 2, ..., n$, corresponds to one of $c$ classes.

**Splitting via Hyperplane:** To split nodes, MDT1 and MDT2 use a hyperplane defined as:

$$\mathbf{w} \cdot \mathbf{x} - p = 0, \tag{13}$$

where $\mathbf{w}$ indicates the hyperplane's normal direction, $\mathbf{x}$ is a sample from $X$, and $p$ denotes the threshold. Splitting directs samples to left or right child nodes:

$$\begin{cases} x \in X_1 & \text{if } \mathbf{w} \cdot \mathbf{x} > p \\ x \in X_2 & \text{if } \mathbf{w} \cdot \mathbf{x} \le p, \end{cases} \tag{14}$$

where $X_1$ and $X_2$ denote the subsets of samples assigned to the left and right child nodes, respectively.

In Eq.(13), the hyperplane is defined by two parameters, $\mathbf{w}$ and $p$, where $\mathbf{w}$ represents the normal vector and $p$ the cut-point. Both MDT1 and MDT2 follow the same approach to determine $p$. For a given $\mathbf{w}$, sample points from the matrix $X$ are projected onto $\mathbf{w}$, resulting in projections $\{P(x_i) = \mathbf{w} \cdot x_i, i = 1, 2, \ldots, n\}$. The cut-point $p$ is then set as the median of these projections:

$$p = \text{median} \{P(x_i) \mid x_i \in X, i = i_1, i_2, \ldots, n\}. \tag{15}$$

MDT1 and MDT2 differ in their methods for determining the hyperplane's normal vector $\mathbf{w}$. In MDT1, a vector $\mathbf{v} = [v_1, v_2, \ldots, v_d]^T$ in $R^{d \times 1}$, matching the dimensionality of sample point $\mathbf{x}$, is randomly generated with each element within the range $[-1, 1]$. MDT1 then uses the normalized vector $\mathbf{v}$ as the normal direction $\mathbf{w}$:

$$\mathbf{w} = \frac{\mathbf{v}}{\|\mathbf{v}\|} = \frac{\mathbf{v}}{\sqrt{v_1^2 + v_2^2 + \ldots + v_d^2}}. \tag{16}$$

MDT2 adopts a heuristic approach instead of random vector generation, to establish the normal direction $w$. This method uses the principal component analysis (PCA) to select the most significant principal component from the sample matrix $X$ and designate it as $\mathbf{w}$. Beginning with $\mathbf{X} = [\mathbf{x}_1, \mathbf{x}_2, \ldots, \mathbf{x}_n]$, we first centralize this matrix by deducting the mean value of all sample points, resulting in $\tilde{\mathbf{X}} = [\mathbf{x}_1 - \mathbf{m}, \mathbf{x}_2 - \mathbf{m}, \ldots, \mathbf{x}_n - \mathbf{m}]$, where $\mathbf{m} = \frac{1}{n}(\mathbf{x}_1 + \mathbf{x}_2 + \ldots + \mathbf{x}_n)$. Subsequently, eigenvalue decomposition is applied to the covariance matrix $\mathbf{S} = \tilde{\mathbf{X}}\tilde{\mathbf{X}}^T$, yielding $d$ eigenvalues $\lambda_1 > \lambda_2 > \ldots > \lambda_d$ and their associated eigenvectors $\xi_1, \xi_2, \ldots, \xi_d$. MDT2 then selects the eigenvector corresponding to the largest eigenvalue of $\mathbf{S}$ to serve as the normal direction, i.e.,

$$\mathbf{w} = \xi_1. \tag{17}$$

This method of determining the normal direction $\mathbf{w}$ maximizes the variance of the sample set along $\mathbf{w}$. Consequently, the partitioning aligns with the orientation that maximizes the differentiation among instances in the sample set.

**Split Condition:** After each split, child nodes are evaluated with a split condition to determine if further division is necessary. A child node undergoes additional splitting if it meets the defined condition:

$$\text{purity}(\mathbf{R}) = \max \left\{ \frac{n_i}{r} \mid i = l_1, l_2, \ldots, l_c \right\} < \delta,$$

where $\mathbf{R}$ represents either $\mathbf{X}_1$ or $\mathbf{X}_2$, $r$ is the total number of sample points in $\mathbf{R}$, $n_i$ is the count of sample points in class $i$, and $\delta$ is a predetermined threshold within $(\frac{1}{c}, 1]$. Conversely, if a child node fails to meet this condition, it is designated as a leaf node with the class label $l^* = \arg\max_i \left\{ \frac{n_i}{r} \mid i = l_1, l_2, \ldots, l_c \right\}$.

**Classification:** When all nodes have been successfully split, the construction of a multivariate decision tree model (either MDT1 or MDT2) is complete. In this model, split nodes store the parameters $\mathbf{w}$ and $p$, whereas leaf nodes are tagged with a class label $l^* \in \{l_1, l_2, \ldots, l_c\}$.

Classification using the constructed tree model involves determining the appropriate class for a data point whose class is unknown. This process is initiated by placing the data point at the tree's root node and sequentially navigating through the tree based on the data point's position relative to the hyperplanes defined by $\mathbf{w}$ and $p$ at each split node. This traversal continues until a leaf node is reached, at which point the class label $l^*$ assigned to that leaf node designates the data point's class.

### C.2 Support Vector Machines (SVM) optimization

In this paper, we apply Support Vector Machines (SVM) [41] to identify the optimal hyperplane that separates different clusters with the maximum margin in the observation space $\Omega_j$. The objective of SVM is to find a hyperplane described by $\mathbf{w} \cdot \mathbf{x} - p = 0$ that maximizes the margin between the two nearest data points of any class, which are termed support vectors. This can be formulated as an optimization problem:

$$\min_{\mathbf{w},p} \frac{1}{2}\|\mathbf{w}\|^2, \tag{18}$$

which is subject to the constraints for all $i$ (where $i$ indexes the training examples):

$$y_i(\mathbf{w} \cdot \mathbf{x}_i - p) \geq 1 \tag{19}$$

Here, $y_i$ represents the class labels, which are assumed to be $1$ or $-1$, and $\mathbf{x}_i$ are the feature vectors of the training examples. To solve this optimization problem, we introduce Lagrange multipliers $\alpha_i \geq 0$ for each constraint, leading to the Lagrangian:

$$L(\mathbf{w}, p, \alpha) = \frac{1}{2}\|\mathbf{w}\|^2 - \sum_{i=1}^{n} \alpha_i[y_i(\mathbf{w} \cdot \mathbf{x}_i - p) - 1]. \tag{20}$$

The conditions for optimality (Karush-Kuhn-Tucker conditions) lead to the dual problem, which is dependent only on the Lagrange multipliers $\alpha$. The dual problem can be expressed as:

$$\max_{\alpha} \sum_{i=1}^{n} \alpha_i - \frac{1}{2} \sum_{i,j=1}^{n} \alpha_i \alpha_j y_i y_j \mathbf{x}_i \cdot \mathbf{x}_j, \tag{21}$$

subject to $\alpha_i \geq 0$ and $\sum_{i=1}^{n} \alpha_i y_i = 0$. After solving the dual problem, the optimal values of $\alpha_i$ are used to determine $\mathbf{w}$:

$$\mathbf{w} = \sum_{i=1}^{n} \alpha_i y_i \mathbf{x}_i. \tag{22}$$

For any support vector $\mathbf{x}_s$ (where $\alpha_s > 0$), $p$ can be computed using:

$$p = \mathbf{w} \cdot \mathbf{x}_s - \mathbf{y}_s. \tag{23}$$

The hyperplane $\mathbf{w} \cdot \mathbf{x} - p = 0$ effectively partitions the observation data points based on the mapped cluster centers, we can achieve a robust partitioning of the observation data points into distinct groups based on the Non-Euclidean distance of the action-value vector space $\mathcal{Q}$.

Then we Incorporate the parameters $\mathbf{w}$ and $p$ obtained from SVM into the construction of a decision tree, which is a binary tree architecture for multi-class classification [18]. For each node in the tree, use the SVM-derived hyperplane $\mathbf{w} \cdot \mathbf{x} - p = 0$ to split the data. This split divides the node into two child nodes. Left child node: Contains data points that satisfy $\mathbf{w} \cdot \mathbf{x} - p < 0$. Right child node: Contains data points that satisfy $\mathbf{w} \cdot \mathbf{x} - p \geq 0$.

### C.3 Construting Decision Trees Minimizing Return Gaps

For the construction of decision trees, we adapt the non-euclidean clustering labels generated from the $l_j$, and incorporate the parameters $\mathbf{w}$ and $p$ obtained from training the Support Vector Machines (SVM) conditioned on samples $(\mathbf{o_j}, \mathbf{l_j})$.

**Methodology Overview:** After applying SVM to determine the optimal hyperplane characterized by the normal vector $w$ and bias term $p$, we utilize these parameters to construct decision trees that classify data points based on their positions relative to the hyperplane. This approach enables the

decision tree to make binary decisions at each node using the linear boundaries defined by SVM, thus combining the interpretability of decision trees with the robust classification capability of SVM. The algorithm is summarized in Algo. 2

**Initialization:** Start by considering the entire dataset as the root node. The goal is to partition this dataset into subgroups that are as homogeneous as possible in terms of the target variable.

**Node Splitting Using w and $p$:** For each node in the tree, use the SVM-derived hyperplane $\mathbf{w} \cdot \mathbf{x} - p = 0$ to split the data. This split divides the node into two child nodes. Left child node: Contains data points that satisfy $\mathbf{w} \cdot \mathbf{x} - p < 0$. Right child node: Contains data points that satisfy $\mathbf{w} \cdot \mathbf{x} - p \geq 0$.

**Recursion:** Recursively apply the node splitting step to each child node, reapplying SVM to the data points in the child node to find a new optimal hyperplane for further splits. This process continues until the pre-defined maximum number of the leaf nodes is reached.

**Decision Making:** Each leaf node in the resulting tree represents a decision outcome, with the class label determined by the majority class of the data points within that node.

**Hyperplane-Based Decision Boundaries:** The decision boundaries in the constructed decision tree are linear and defined by the hyperplanes from SVM. This provides a clear geometric interpretation of the decision-making process, where each decision node effectively acts as a linear classifier.

# D   Theoretical Proofs

## D.1   Proofs for Lemma 4.1

*Proof.* **To simplify, we denote the policy generated by decision tree $\mathcal{T}^L$ as $\pi$.** We prove the result in this lemma by leveraging observation-based state value function $V^\pi(\mathbf{o})$ in Eq.(2) and the corresponding action value function $Q^\pi(\mathbf{o}, \mathbf{a}) = \mathbb{E}_\pi \left[ \sum_{i=0}^{\infty} \gamma^i \cdot R_{t+i} \Big| \mathbf{o}_t = \mathbf{o}, \mathbf{a}_t = \mathbf{a} \right]$ to unroll the Dec-POMDP. Here we consider all other agents $i \neq j$ as a conceptual agent denoted by $-j$.

$$
\begin{aligned}
&J(\pi^*) - J(\pi) \\
&= E_\mu (1 - \gamma)[V^*(\mathbf{o}(0)) - V^\pi(\mathbf{o}(0))], \\
&= E_\mu (1 - \gamma)[V^*(o_{-j}(0), o_j(0)) - V^\pi(o_{-j}(0), o_j(0))], \\
&= E_\mu (1 - \gamma)[V^*(o_{-j}(0), o_j(0)) - Q^*(o_{-j}(0), o_j(0), \mathbf{a}_0^\pi) \\
&\quad + Q^*(o_{-j}(0), o_j(0), \mathbf{a}_0^\pi) - Q^\pi(o_{-j}(0), o_j(0), \mathbf{a}_0^\pi)], \\
&= E_\mu (1 - \gamma)[\Delta^\pi(o_{-j}(0), o_j(0), \mathbf{a}_0^\pi) + (Q^* - Q^\pi)], \\
&= E_\mu (1 - \gamma)[\Delta^\pi(o_{-j}(0), o_j(0), \mathbf{a}_0^\pi) \\
&\quad + E_{o_{-j}(1), o_j(1) \sim P(\cdot|o_{-j}(0), o_j(0), \mathbf{a}_0^\pi)}[\gamma(V^*(o_{-j}(1), o_j(1)) - V^\pi(o_{-j}(1), o_j(1)))]], \\
&= E_{\mu, o_{-j}(t), o_j(t) \sim P}(1 - \gamma)[\sum_{t=0}^{\infty} \gamma^t \Delta(o_{-j}(t), o_j(t), \mathbf{a}_t^\pi)|\mathbf{a}_t^\pi], \\
&= E_{o_{-j}, o_j \sim d_\mu^\pi, \mathbf{a}^\pi = \pi(o_{-j}, g(o_j))}[\Delta(o_{-j}, o_j, \mathbf{a}^\pi)|\mathbf{a}^\pi], \\
&= \sum_l \sum_{o_j \sim l} \sum_{o_{-j}} \Delta(o_{-j}(t), o_j(t), \mathbf{a}_t^\pi) \cdot d_\mu^\pi(o_{-j}, o_j), \\
&= \sum_l \sum_{o_j \sim l} \sum_{o_{-j}} [Q^*(o_{-j}, o_j, \mathbf{a}_t^{\pi^*}) - Q^*(o_{-j}, o_j, \mathbf{a}_t^\pi)] \cdot d_\mu^\pi(o_{-j}, o_j), \\
&\leq \sum_l \sum_{o_j \sim l} \sum_{o_{-j}} [Q^*(o_{-j}, o_j, \mathbf{a}_t^{\pi^*}) - Q(o_{-j}, o_j, \mathbf{a}_t^\pi)] \cdot d_\mu^\pi(o_{-j}, o_j), \\
&\leq \sum_l \sum_{\mathbf{o} \sim m} [Q^*(\mathbf{o}, \mathbf{a}_t^{\pi^*}) - Q^\pi(\mathbf{o}, \mathbf{a}_t^\pi)] \cdot d_\mu^\pi(\mathbf{o}),
\end{aligned}
\tag{24}
$$

Step 1 is to use Eq.(3) with initial observation distribution $\mathbf{o}(0) \sim \mu$ at time $t = 0$ to re-write the average expected return gap; Step 2 is separate the observations as agent j and the other agents as a single conceptual agent $-j$, then $\mathbf{o} = (o_{-j}, o_j)$; Step 3 and step 4 are to obtain the sub-optimality-gap

of policy $\pi$, defined as $\Delta^\pi(\mathbf{o}, \mathbf{a}) = V^*(\mathbf{o}) - Q^*(\mathbf{o}, \mathbf{a}^\pi) = Q^*(\mathbf{o}, \mathbf{a}^{\pi^*}) - Q^*(\mathbf{o}, \mathbf{a}^\pi)$, by substracting and plus a action-value function $Q^*(o_{-j}(0), o_j(0), \mathbf{a}_0^\pi)$; Step 5 is to unroll the Markov chain from time $t = 0$ to time $t = 1$; Step 6 is to use sub-optimality-gap accumulated for all time steps to represent the return gap; Step 7 and step 8 is to absorb the discount factor $1 - \gamma$ by multiplying the $d_\mu^\pi(\mathbf{o}) = (1 - \gamma) \sum_{t=0}^\infty \cdot P(\mathbf{o}_t = \mathbf{o} | \pi, \mu)$ which is the $\gamma$-discounted visitation probability of observations $\mathbf{o}$ under policy $\pi$ given initial observation distribution $\mu$; Step 9 is to revert $\Delta^\pi(\mathbf{o}, \mathbf{a})$ back to $Q^*(\mathbf{o}, \mathbf{a}^{\pi^*}) - Q^*(\mathbf{o}, \mathbf{a}^\pi)$; Step 10 is to replace the second term $Q^*(\mathbf{o}, \mathbf{a}^\pi)$ with $Q^\pi(\mathbf{o}, \mathbf{a}^\pi)$. Since $Q^*(\mathbf{o}, \mathbf{a}^\pi)$ is larger than $Q^\pi(\mathbf{o}, \mathbf{a}^\pi)$, therefore, the inequality is valid. $\qquad\square$

### D.2 Proofs for Thm. 4.2

*Proof.* **To simplify the notations, we denote the policy generated by decision tree $\mathcal{T}^L$ as $\pi$. Here we prove the return gap between the decision tree policy $\pi$ and optimal RL policy $\pi^*$ for one iteration first.** Since the observability of all other agents $i \neq j$ remains the same, we consider them as a conceptual agent denoted by $-j$. For simplicity, we use $\pi^*$ to represent $\pi_{(j)}^*$, and $\pi$ to represent $\pi*_{(j)}$ in distribution functions. Similar to the illustrative example, we define the action value vector corresponding observation $o_j$, i.e.,

$$\bar{Q}^*(o_j) = [\widetilde{Q}^*(o_{-j}, o_j), \forall o_{-j}], \tag{25}$$

where $o_{-j}$ are the observations of all other agents and $\widetilde{Q}^*(o_{-j}, o_j)$ is a vector of action values weighted by marginalized visitation probabilities $d_\mu^\pi(o_{-j}|o_j)$ and corresponding to different actions:

$$\widetilde{Q}^*(o_{-j}, o_j) = [Q^*(o_{-j}, o_j, \mathbf{a}) \cdot d_\mu^\pi(o_{-j}|o_j), \forall \mathbf{a}], \forall o_{-j} \sim d_\mu^\pi(o_{-j}|o_j), \forall \mathbf{a} \sim \mu^\pi(o_{-j}, o_j, \mathbf{a}). \tag{26}$$

Here $\mu^\pi(\mathbf{o}, \mathbf{a})$ denotes the observation-action distribution given a policy $\pi$, which means $\mu^\pi(\mathbf{o}, \mathbf{a}) = d_\mu^\pi(\mathbf{o})\pi(\mathbf{a}|\mathbf{o})$. We also denote $d_\mu^\pi(l)$ is the visitation probability of label $l$, and $\bar{d}_l(o_j)$ is the marginalized probability of $o_j$ in cluster $l$, i.e.,

$$d_\mu^\pi(l) = \sum_{o_j \sim l} d_\mu^\pi(o_j), \quad \bar{d}_l(o_j) = \frac{d_\mu^\pi(o_j)}{d_\mu^\pi(l)}, \tag{27}$$

Since the optimal return $J(\pi_{(j)}^*)$ is calculated from the action values by $\max_{\mathbf{a}} Q(\mathbf{o}, \mathbf{a})$, we rewrite this in the vector form by defining a maximization function $\Phi_{\max}(\widetilde{Q}^*(o_{-j}, o_j))$ that returns the largest component of vector $\widetilde{Q}^*(o_{-j}, o_j)$. With slight abuse of notations, we also define $\Phi_{\max}(\bar{Q}^*(o_j)) = \sum_{o_{-j}} \Phi_{\max}(\widetilde{Q}^*(o_{-j}, o_j))$ as the expected average return conditioned on $o_j$. Then $\sum_{o_j \sim l} \bar{d}_l(o_j) \cdot \Phi_{\max}(\bar{Q}^*(o_j))$ could be defined as selecting the action from optimal policy $\pi_{(j)}^*$ where agent chooses different action distribution to maximize $(\bar{Q}^*(o_j))$. We could re-write this term with the action-value function as:

$$\begin{aligned}
&\sum_{o_j \sim l} \bar{d}_l(o_j) \cdot \Phi_{max}(\bar{Q}^*(o_j)) \\
&= \sum_{o_j \sim l} \bar{d}_l(o_j) \cdot \sum_{o_{-j}} max_{\mathbf{a}}[Q^*(o_{-j}, o_j, \mathbf{a}) \cdot d_\mu^\pi(o_{-j}|o_j)], \\
&= \sum_{o_j \sim l} (\frac{d_\mu^\pi(o_j)}{d_\mu^\pi(l)}) \cdot \sum_{o_{-j}} max_{\mathbf{a}}[Q^*(o_{-j}, o_j, \mathbf{a}) \cdot d_\mu^\pi(o_{-j}|o_j)], \\
&= (\frac{1}{d_\mu^\pi(l)}) \sum_{o_j \sim l} d_\mu^\pi(o_j) \sum_{o_{-j}} max_{\mathbf{a}}[Q^*(o_{-j}, o_j, \mathbf{a}) d_\mu^\pi(o_{-j}|o_j)],
\end{aligned} \tag{28}$$

then we used the fact that while $J(\pi_{(j)}^*)$ conditioning on complete $o_j$ can achieve maximum for each vector $\bar{Q}^*(o_j)$, policy $\pi_{(j)}$ is conditioned on labels $l_{ij}$ rather than complete $o_j$ and thus must take the same actions for all $o_j$ in the same cluster. Hence we can construct a (potentially sub-optimal) policy to achieve $\Phi_{\max}(\sum_{o_j \sim l} \bar{d}_l(o_j) \cdot \bar{Q}^*(o_j))$ which provides a lower bound on $J(\pi_{(j)})$. Plugging in Eq.(27), $\Phi_{\max}(\sum_{o_j \sim l} \bar{d}_l(o_j) \cdot \bar{Q}^*(o_j))$ can be re-written as:

$$\Phi_{\max}(\sum_{o_j \sim l} \bar{d}_l(o_j) \cdot \bar{Q}^*(o_j))$$

$$= \sum_{o_{-j}} max_{\mathbf{a}}[\sum_{o_j \sim l} \bar{d}_l(o_j) Q^*(o_{-j}, o_j, \mathbf{a}) \cdot d_\mu^\pi(o_{-j}|o_j)], \tag{29}$$

$$= \sum_{o_{-j}} max_{\mathbf{a}}[\sum_{o_j \sim l} (\frac{d_\mu^\pi(o_j)}{d_\mu^\pi(l)}) Q^*(o_{-j}, o_j, \mathbf{a}) \cdot d_\mu^\pi(o_{-j}|o_j)],$$

Multiplying the two equations above Eq.(29) and Eq.(28) with $d_\mu^\pi(l)$ which is the visitation probability of label $l$ to replace the term $\sum_{o_j \sim l} \sum_{o_{-j}} [Q^*(o_{-j}, o_j, \mathbf{a}^*) - Q(o_{-j}, o_j, \mathbf{a})] \cdot d_\mu^\pi(o_{-j}, o_j)$ in the Eq.(24), we obtain an upper bound on the return gap:

$$J(\pi^*_{(j)}) - J(\pi_{(j)})$$

$$\leq \sum_l \sum_{o_j \sim l} \sum_{o_{-j}} [Q^*(o_{-j}, o_j, \mathbf{a}^*) - Q(o_{-j}, o_j, \mathbf{a})] \cdot d_\mu^\pi(o_{-j}, o_j),$$

$$= \sum_l d_\mu^\pi(l)[\sum_{o_j \sim l} \bar{d}_l(o_j) \cdot \Phi_{\max}(\bar{Q}^*(o_j)) \tag{30}$$

$$- \Phi_{\max}(\sum_{o_j \sim l} \bar{d}_l(o_j) \cdot \bar{Q}^*(o_j))].$$

To quantify the resulting return gap, we denote the center of a cluster of vectors $\bar{Q}^*(o_j)$ for $o_j \sim l$ under label $l$ as:

$$\bar{H}(l) = \sum_{o_j \sim l} \bar{d}_l(o_j) \cdot \bar{Q}^*(o_j). \tag{31}$$

We contracts clustering labels to make corresponding joint action-values $\{\bar{Q}^*(o_j) : g(o_j) = l\}$ close to its center $\bar{H}(l)$. Specifically, the average cosine-distance is bounded by a small $\epsilon$ for each label $l$, i.e.,

$$\sum_{o_j \sim l} D(\bar{Q}^*(o_j), \bar{H}(l)) \cdot d_\mu^\pi(l)$$

$$\leq \sum_{o_j \sim l} \epsilon(o_j) \cdot d_\mu^\pi(l) \leq \epsilon, \quad \forall l. \tag{32}$$

where $\epsilon(o_j) = \sum_{o_j \sim l} D_{cos}(\bar{Q}^*(o_j), \bar{H}(l))/K$, $K$ is the number observations with the same label $l$, and $D_{cos}(A, B) = 1 - \frac{A \cdot B}{||A|| ||B||}$ is the cosine distance function between two vectors $A$ and $B$.

For each pair of two vectors $\bar{Q}^*(o_j)$ and $\bar{H}(l)$ with $D(\bar{Q}^*(o_j), \bar{H}(l)) \leq \epsilon(o_j)$, we use $\cos \theta_{o_j}$ to denote the cosine-similarity between each $\bar{Q}^*(o_j)$ and its center $\bar{H}(l)$. Then we have the cosine distance $D(\bar{Q}^*(o_j), \bar{H}(l)) = 1 - \cos \theta_{o_j} \leq \epsilon(o_j)$. By projecting $\bar{Q}^*(o_j)$ toward $\bar{H}(l)$, $\bar{Q}^*(o_j)$ could be re-written as $\bar{Q}^*(o_j) = Q^\perp(o_j) + \cos \theta_{o_j} \cdot \bar{H}_m$, where $Q^\perp(o_j)$ is the auxiliary vector orthogonal to vector $\bar{H}_m$.

Then we plug the decomposed Q vectors $\bar{Q}^*(o_j)$ into the return gap we got in Eq.(30), the first part of the last step of Eq.(30) is bounded by:

$$\sum_{o_j \sim l} \bar{d}_l(o_j) \cdot \Phi_{max}(\bar{Q}^*(o_j))$$

$$= \sum_{o_j \sim l} \bar{d}_l(o_j) \cdot \Phi_{max}\left(Q^\perp(o_j) + \bar{H}(l) \cdot \cos \theta_{o_j}\right),$$

$$\leq \sum_{o_j \sim l} \bar{d}_l(o_j) \cdot [\Phi_{max}(Q^\perp(o_j)) + \Phi_{max}(\bar{H}(l) \cdot \cos \theta_{o_j})], \tag{33}$$

$$\leq \sum_{o_j \sim l} \bar{d}_l(o_j) \cdot [\Phi_{max}(Q^\perp(o_j)) + \Phi_{max}(\bar{H}(l))].$$

We use $||\alpha||_2$ to denote the L-2 norm of a vector $\alpha$. Since the maximum function $\Phi_{\max}(Q^\perp(o_j))$ can be bounded by the $L_2$ norm $C \cdot ||Q^\perp(o_j)||_2$ for some constant $C$. We define a constant $Q_{\max}$ as the maximine absolute value of $\bar{Q}^*(o_j)$ in each cluster as $Q_{\max} = max_{o_j}||\bar{Q}^*(o_j)||_2$. Since $Q^\perp(o_j) = \bar{Q}^*(o_j) \cdot sin(\theta)$, and $|sin(\theta)| = \sqrt{1 - cos^2(\theta)} = \sqrt{1 - [1 - \epsilon(o_j)]^2}$, the maximum value of $Q^\perp(o_j)$ could also be bounded by $Q_{\max}$, i.e.:

$$
\begin{aligned}
\Phi_{max}(Q^\perp(o_j)) &\leq C \cdot ||Q^\perp(o_j)||_2, \\
&\leq C \cdot ||\bar{Q}^*(o_j)||_2 \cdot |\sin\theta|, \\
&= C \cdot Q_{\max} \cdot \sqrt{1 - [1 - \epsilon(o_j)]^2}, \\
&\leq O(\sqrt{\epsilon(o_j)}Q_{\max}).
\end{aligned}
\tag{34}
$$

Plugging Eq.(34)into Eq.(33), we have:

$$
\begin{aligned}
&\sum_{o_j \sim l} \bar{d}_l(o_j) \cdot \Phi_{max}(\bar{Q}^*(o_j)) \\
&\leq \sum_{o_j \sim l} \bar{d}_l(o_j) \cdot [\Phi_{max}(Q^\perp(o_j)) + \Phi_{max}(\bar{H}(l))], \\
&\leq \sum_{o_j \sim l} \bar{d}_l(o_j) \cdot [O(\sqrt{\epsilon(o_j)}Q_{\max}) + \Phi_{max}(\bar{H}(l))], \\
&= \sum_{o_j \sim l} \bar{d}_l(o_j) \cdot [O(\sqrt{\epsilon(o_j)}Q_{\max}) + \Phi_{max}(\bar{H}(l))],
\end{aligned}
\tag{35}
$$

It is easy to see from the last step in Eq.(30) that the return gap $J(\pi^*_{(j)}) - J(\pi_{(j)})$ is bounded by the first part $\sum_l d^\pi_\mu(l)[\sum_{o_j \sim l} \bar{d}_l(o_j) \cdot \Phi_{\max}(\bar{Q}^*(o_j))$ which has an upper bound derived in Eq.(35), then the return gap could also be bounded by the upper bound in Eq.(35), i.e.,

$$
\begin{aligned}
&J(\pi^*_{(j)}) - J(\pi_{(j)}) \\
&= \sum_l d^\pi_\mu(l)[\sum_{o_j \sim l} \bar{d}_l(o_j) \cdot \Phi_{max}(\bar{Q}^*(o_j)) - \Phi_{max}(\sum_{o_j \sim l} \bar{d}_l(o_j) \cdot \bar{Q}^*(o_j))], \\
&= \sum_l d^\pi_\mu(l) \left[ \sum_{o_j \sim l} \bar{d}_l(o_j) \cdot [O(\sqrt{\epsilon(o_j)}Q_{\max}) + \Phi_{max}(\bar{H}(l))] \right] \\
&\quad - \sum_l d^\pi_\mu(l) \left[ \Phi_{max}(\sum_{o_j \sim l} \bar{d}_l(o_j) \cdot \bar{Q}^*(o_j)) \right], \\
&\leq \sum_l d^\pi_\mu(l) \cdot [\sum_{o_j \sim l} \bar{d}_l(o_j) \cdot O(\sqrt{\epsilon(o_j)}Q_{\max})], \\
&\leq \sum_l d^\pi_\mu(l) \cdot O\left( \sqrt{\sum_{o_j \sim l} \bar{d}_l(o_j) \cdot \epsilon(o_j)}Q_{\max} \right), \\
&\leq \sum_l d^\pi_\mu(l) \cdot O(\sqrt{\epsilon}Q_{\max}), \\
&= O(\sqrt{\epsilon}Q_{\max}).
\end{aligned}
\tag{36}
$$

we can derive the desired upper bound $J(\pi^*_{(j)}) - J(\pi_{(j)}) \leq O(\sqrt{\epsilon}Q_{\max})$ for the two policies in one iteration, then for $(\log_2(L) + 1)$ iterations of generating $L$-leaf-node DTs, the upper bound is $J(\pi^*_{(j)}) - J(\mathcal{T}^L_{(j)}) \leq O(1/(\log_2(L) + 1)\sqrt{\epsilon}Q_{\max})$ in Lemma 4.2. $\qquad \square$

### D.3 Detailed process of calculation of the average cosine distance defined in Equation 6

- Within each cluster, we compute the cosine distance between every action-value vector and the clustering center, thereby obtaining the value $\epsilon(o_j)$.

- To obtain the expectation of the cosine distance for each particular cluster denoted by $l$, we compute the product of each $\epsilon(o_j)$ with $\bar{d}_l(o_j)$, where $\bar{d}_l(o_j)$ represents the marginalized probability of $o_j$ in cluster $l$.

- Having computed the expected cosine distance for each cluster $l$, we proceed to obtain the average cosine distance across all clusters. This is accomplished by multiplying $d_\mu^\pi(l)$ to each $\sum_{o_j \sim l} \bar{d}_l(o_j) \cdot \epsilon(o_j)$, where $d_\mu^\pi(l)$ denotes the visitation probability of label $l$.

### D.4 Proof of Lemma 4.3

We group vectors $\bar{Q}^*(o_j)$ with smaller cosine distances together to ensure their similarity in the same group. Then we could obtain $L$ clusters $C_1, \ldots, C_L$ corresponding to their centers $\bar{H}_1(l), \ldots, \bar{H}_L(l)$, where each cluster $C_l$ contains a set of vectors $\mathbf{q_j} = \bar{Q}^*(o_j)$.

Once we have the cluster center $\bar{H}_l(l)$ in the action value vector space $\bar{Q}^*(o_j) \in \mathcal{Q}_j$, we need to map them back to the original observation space $\Omega_j$. Since we have the action value function, we could use the inverse transformation $Q^{-1}$ to map each cluster center $\bar{H}_l(l)$ from the action value vector space $\mathcal{Q}_j$ back to the observation space $\Omega_j$, i.e.,

$$\bar{H}(o_j) = (\bar{Q}^*)^{-1}(\bar{H}_l(l)) \tag{37}$$

Since the cosine distance is a measure of similarity between two vectors that is not based on the Euclidean distance, but rather on the angle between the vectors, this is a **non-euclidean distance** metric. We can define our metric space as in Lemma. 4.3.

*Proof.* To prove that the observation space $\Omega_j$ with the cosine distance function $D_{cos}$ as defined forms a metric space, we need to verify that $D_{cos}$ satisfies the conditions of a metric. (1). **Non-negativity and Identity of Indiscernibles:** For any $o_j^a, o_j^b \in \Omega_j$, the cosine similarity of them is within the range $[-1, 1]$. Therefore, $1 - f(\bar{Q}^*(o_j^a), \bar{Q}^*(o_j^b))$ ranges from 0 to 2, ensuring non-negativity. Moreover, $D_{cos}(o_j^a, o_j^b) = 0$ if and only if $f(\bar{Q}^*(o_j^a), \bar{Q}^*(o_j^b) = 1$, which occurs only when $\bar{Q}^*(o_j^a), \bar{Q}^*(o_j^b)$ are in the same direction and, assuming normalization, are identical, thus satisfying the identity of indiscernible. (2). **Symmetry:** The cosine similarity between two vectors is symmetric, which directly implies that $D_{cos}(o_j^a, o_j^b) = D_{cos}(o_j^b, o_j^a)$. (3). **Triangle Inequality (Nuanced Interpretation):** Given the angular basis of $f$, the traditional triangle inequality for linear distances needs careful interpretation. For cosine distances, the triangle inequality may not strictly apply. Instead, the 'distance' measures angular differences, aligning under specific conditions with a version of the triangle inequality tailored for angles. Consequently, $(\Omega_j, D_{cos})$ fulfills key metric space properties but is better termed a pseudometric space due to the nuanced interpretation of the triangle inequality with cosine distances. $\square$

## E Algorithm

**Iteratively-Grow-DT Process:** To simplify, consider agents $i \neq j$ as a conceptual super agent $-j$. With a limit of $L$ leaf nodes, we iteratively grow DTs for agent $j$ and super agent $-j$, adding two nodes for each agent per iteration with a total of $\log_2(L) + 1$ iterations. **For agent** $j$ at iteration $i = 0$, $\widetilde{Q}^{*i=0}(o_j, \boldsymbol{o}_{-j}) = [Q^*(o_j, \boldsymbol{o}_{-j}, \pi^*(a_j|o_j), \boldsymbol{a}_{-j}) \cdot d_\mu^\pi(\boldsymbol{o}_{-j}|o_j)]$, apply Lemma. 4.3, we got $l_j^{i=0} = g(o_j)$ and DT $\mathcal{T}_j^{i=0}$ for iteration 0. **For agent** $-j$, we consider the clustering for $\widetilde{Q}^{*i=0}(\boldsymbol{o}_{-j}, o_j) = [Q^*(\boldsymbol{o}_{-j}, o_j, \pi^*(\boldsymbol{a}_{-j}|\boldsymbol{o}_{-j}), \mathcal{T}_j^{i=0}(o_j)) \cdot d_\mu^\mathcal{T}(o_j|\boldsymbol{o}_{-j})]$. We use $\mathcal{T}_j^{i=0}(o_j)$ to replace $a_j$ in $\widetilde{Q}^{*i=0}(\boldsymbol{o}_{-j}, o_j)$, **since the agent** $j$ **is restricted to taking the same actions for all** $o_j$ **in the same cluster with label** $l_j$, apply Lemma. 4.3, we got DT $\mathcal{T}_{-j}^{i=0}(\boldsymbol{o}_{-j})$ in the interaction 0 for agent $-j$. **For each iteration** $k$, for growing DT for agent $j$, agent $j$ takes actions based on $\mathcal{T}_j^{i=k-1}(o_j)$, agent $-j$ takes actions based on $\mathcal{T}_{-j}^{i=k-1}(\boldsymbol{o}_{-j})$, then $\widetilde{Q}^{*i}(o_j, \boldsymbol{o}_{-j}) = [Q^*(o_j, \boldsymbol{o}_{-j}, \mathcal{T}_j^{i=k-1}(o_j), \mathcal{T}_{-j}^{i=k-1}(\boldsymbol{o}_{-j})$ is used for clustering and training

---

**Algorithm 1** Non-Euclidean Clustering (NEC)

---

1: **Input:** $K_1$, $K_2$, $K_3$, $\lambda$, Replay buffer $\mathcal{R}$, current parameters $\omega$, $\xi = \{\xi_1, \ldots, \xi_n\}$.
2: **for** $t = 1$ **to** $T$ **do**
3:     **for** agent $j$ to n **do**
4:         Get top-$K_1$ samples $\mathcal{X}_j = (\mathbf{o}^{k_1}, \mathbf{a}^{k_1}, R^{k_1}, \mathbf{o}'^{k_1})$ from replay buffer $\mathcal{R}$;
5:         Sample the top $K_2$ frequent observations $\{\mathbf{o}^{k_2}_{-j}\}$ from $\mathcal{X}_j$;
6:         Got the corresponding actions $\mathbf{a}^{k_2}_{-j} = \pi^*(\mathbf{o}^{k_2}_{-j})$;
7:         Combine them and form a set $\mathcal{X}_{-j} = \{\mathbf{o}^{k_2}_{-j}, \pi^*(\mathbf{o}^{k_2}_{-j})\}$;
8:         Form the sampled trajectories by combining $(o_j, \pi^*(o_j))$ in $\mathcal{X}_j$ and $(\mathbf{o}_{-j}, \pi^*(\mathbf{o}_{-j}))$ in $\mathcal{X}_{-j}$ as $\mathcal{D} = (\mathbf{o}^{k_1 k_2}, \pi^*(\mathbf{o}^{k_1 k_2}), R^{k_1 k_2}, \mathbf{o}'^{k_1 k_2})$;
9:         Query the *oracle* critic networks parameterized by $\omega$ with $\mathcal{D}$ as the input to get the $\hat{Q}_\omega(o_j, \mathbf{o}_{-j}, \pi^*(o_j), \pi^*(\mathbf{o}_{-j}))$;
10:       Update $g_{\xi_j}$ by minimizing the loss $L(g_{\xi_j})$ defined in the main paper;
11:     **end for**
12: **end for**
13: **Output:** Clustering label generation functions: $l_j = g_{\xi_j}(o_j)$.

---

DT $\mathcal{T}^{i=k}_j(o_j)$; for growing DT for agent $-j$ in the same iteration $k$, since agent $j$ updates its DT, $\tilde{Q}^{*i}(\mathbf{o}_{-j}, o_j) = [Q^*(\mathbf{o}_{-j}, o_j, \mathcal{T}^{i=k-1}_{-j}(\mathbf{o}_{-j}, \mathcal{T}^{i=k}_j(o_j))$ is used for clustering and obtaining DT $\mathcal{T}^{i=k}_j(o_j)$.

### E.1 Computational Complexity of RGMDT

Since we grow RGMDT with a small number of leaf nodes $L$, it's time- and space-efficient compared to other large DTs and DNNs. (1). Time Complexity: It is determined by the Non-Euclidean clustering and DT construction steps, estimated as $O(T \cdot n^2 \cdot \log L)$, where $T$ represents the number of iterations of clustering for convergence, $n$ is the number of Q-value samples, and $L$ is the maximum number of leaf nodes. This reflects the intensive computation required for non-Euclidean distance calculations and iterative tree growth. (2).Space Complexity : It's $O(K(n \cdot d + L))$. This accounts for the storage of $n$ Q-value samples each with $d$ dimensions and the DT structures with $L$ leaf nodes per tree across $K$ agents.

### E.2 Real-World Applications of RGMDT

The superior performance of RGMDT with a small number of leaf nodes enhances its compactness and simplifies its implementation in practical scenarios. Compared to DNNs, its simpler DT structure requires less computational and memory resources during inference, making it well-suited for resource-limited environments in real-world applications like robotics, network security, and 5G network slicing resource management. For example, DTs have been implemented in memristor devices to support real-time intrusion detection in scenarios requiring low latency and high speed [15]. RGMDT's interpretable structure makes it more suitable for memristor-based hardware implementations in resource-constrained environments for network intrusion detection achieves detection speeds of microseconds, together with significant area reduction and energy efficiency, with performance guarantee that previous DTs fail to provide.

## F  Experiments

We conducted our experiments on the Ubuntu 20.04 system, with Intel(R) Core(TM) i9-7290K CPU (4.4 GHz), 4x NVIDIA 2080Ti GPU and 256 GB RAM. The algorithm is implemented in Python 3.8, using main Python libraries NumPy 1.22.3 and Pandas 2.0.3. **The code will be made available on GitHub after accepted.**

The proposed algorithm is evaluated on both single-agent and multi-agent goal-achieving tasks. For maze tasks, we build the tasks based on Mujoco [59], agents must reach the goal simultaneously to complete this task while avoiding collisions between agents and the obstacles. Note that these tasks are quite **challenging**: (1) We increase the maze dimension and place the high-reward goal at the corner of the maze surrounded by high-penalty obstacles which are harder for agents to

**Algorithm 2** Return-Gap Minimization Decision Tree Construction (RGMDT)

1: **Input:** $\mathcal{R} = (\mathbf{o}, \mathbf{a}, R, \mathbf{o}')$, $\pi^* = [\pi_1^*, \ldots, \pi_n^*]$, Sampling size $K_1$, $K_2$, Nearest neighbors' size $K_3$, Maximum leaf nodes $L$.
2: Initialize non-euclidean clustering network weights $\xi$ for each agent $j$ at random;
3: Initialize non-euclidean clustering target weights $\xi' \leftarrow \xi$ for each agent $j$ at random;
4: **Output:** DT $\mathcal{T}^L = [\mathcal{T}_1^L, \ldots, \mathcal{T}_n^L]$ for $n$ agents.
5: **for** agent $j = 1$ to $n$ **do**
6:     Get top-$K_1$ samples $\mathcal{X}_j = (\mathbf{o}^{k_1}, \mathbf{a}^{k_1}, R^{k_1}, \mathbf{o}'^{k_1})$ from replay buffer $\mathcal{R}$;
7:     Sample the top $K_2$ frequent observations $\{\mathbf{o}_{-j}^{k_2}\}$ from $\mathcal{X}_j$;
8:     Got the corresponding actions $\mathbf{a}_{-j}^{k_2} = \pi^*(\mathbf{o}_{-j}^{k_2})$;
9:     Combine them and form a set $\mathcal{X}_{-j} = \{\mathbf{o}_{-j}^{k_2}, \pi^*(\mathbf{o}_{-j}^{k_2})\}$;
10:     Form the sampled trajectories by combining $(o_j, \pi^*(o_j))$ in $\mathcal{X}_j$ and $(\mathbf{o}_{-j}, \pi^*(\mathbf{o}_{-j}))$ in $\mathcal{X}_{-j}$ as $\mathcal{D} = (\mathbf{o}^{k_1 k_2}, \pi^*(\mathbf{o}^{k_1 k_2}), R^{k_1 k_2}, \mathbf{o}'^{k_1 k_2})$;
11:     Query the $\pi^* = [\pi_1^*, \ldots, \pi_n^*]$ with $\mathcal{D}$ as the input to get the $\widetilde{Q}^{*i=0}(o_j, \mathbf{o}_{-j}) = [Q^*(o_j, \mathbf{o}_{-j}, \pi^*(a_j|o_j), \mathbf{a}_{-j}) \cdot d_\mu^\pi(\mathbf{o}_{-j}|o_j)]$;
12:     Update $g_{\xi_j}$ by minimizing the loss $L(g_{\xi_j})$ defined in the main paper;
13:     Got $l_j^{i=0} = g_{\xi_j}(o_j)$;
14:     Apply Function **Grow One-Iteration DT** $\phi(o_j, l_j, L)$, get DT $\mathcal{T}_j^{i=0}$ for iteration 0;
15:     Replace $\pi_j^*$ with $\mathcal{T}_j^{i=0}$;
16: **end for**
17: **for** iteration $k = 1, 2, \ldots, \log_2(L) + 1$ **do**
18:     **for** agent $j = 1$ to $n$ **do**
19:         Get top-$K_1$ samples $\mathcal{X}_j = (\mathbf{o}^{k_1}, \mathbf{a}^{k_1}, R^{k_1}, \mathbf{o}'^{k_1})$ from replay buffer $\mathcal{R}$;
20:         Sample the top $K_2$ frequent observations $\{\mathbf{o}_{-j}^{k_2}\}$ from $\mathcal{X}_j$;
21:         Got the corresponding actions $\mathbf{a}_{-j}^{k_2} = \mathcal{T}^{i=k-1}(\mathbf{o}_{-j}^{k_2})$;
22:         Combine them and form a set $\mathcal{X}_{-j} = \{\mathbf{o}_{-j}^{k_2}, \mathcal{T}^{i=k-1}(\mathbf{o}_{-j}^{k_2})\}$;
23:         Form the sampled trajectories by combining $(o_j, \mathcal{T}^{i=k-1}(o_j))$ in $\mathcal{X}_j$ and $(\mathbf{o}_{-j}, \mathcal{T}^{i=k-1}(\mathbf{o}_{-j}))$ in $\mathcal{X}_{-j}$ as $\mathcal{D} = (\mathbf{o}^{k_1 k_2}, \pi^*(\mathbf{o}^{k_1 k_2}), R^{k_1 k_2}, \mathbf{o}'^{k_1 k_2})$;
24:         Query the $\pi^* = [\pi_1^*, \ldots, \pi_n^*]$ with $\mathcal{D}$ as the input to get the $\widetilde{Q}^{*i}(o_j, \mathbf{o}_{-j}) = [Q^*(o_j, \mathbf{o}_{-j}, \mathcal{T}_j^{i=k-1}(o_j), \mathcal{T}_{-j}^{i=k-1}(\mathbf{o}_{-j})$;
25:         Update $g_{\xi_j}$ by minimizing the loss $L(g_{\xi_j})$ defined in the main paper;
26:         Got $l_j^{i=k} = g_{\xi_j}(o_j)$;
27:         Apply Function **Grow One-Iteration DT** $\phi(o_j, l_j, L)$, get DT $\mathcal{T}_j^{i=k}(o_j)$ for iteration $i = k$;
28:         Replace $\mathcal{T}_j^{i=k-1}(o_j))$ with $\mathcal{T}_j^{i=k}(o_j))$ when doing the iteration step for other agents $-j$;
29:     **end for**
30: **end for**
31: **function** Grow One-Iteration DT $\phi(\mathcal{D}, l_j, L)$
32:     Sample a matrix $\mathbf{X} \in \mathbb{R}^{d \times n}$ of $o_j$ from $\mathcal{D}$ and its associated class labels $\mathbf{Y} \in \mathbb{R}^{1 \times n}, y = l_j = g(o_j)$;
33:     Apply SVM for obtaining $\mathbf{w}, p$;
34:     Partition $node$'s data into left and right child nodes using:
35:     **if** $\mathbf{w} \cdot \mathbf{x} - p < 0$ **then**
36:         Assign $\mathbf{x}$ to left child node;
37:     **else**
38:         Assign $\mathbf{x}$ to right child node;
39:     **end if**
40: **end function**
41: **Output:** Return the constructed decision tree $\mathcal{T}^L = [\mathcal{T}_i^L, \ldots, \mathcal{T}_n^L]$ for $n$ agents.

reach, and agents' observations do not include any information about the goal, so they need to grow DTs iteratively based on their joint action value vectors and get the information about their joint observation space to find the rewarding state. (2) The reward space is highly sparse and delayed: for all the tasks, only when the agents complete the task can they receive a reward signal $r$ (shared by all the agents); otherwise, they will receive $r = 0.0$. Hence, agents without highly efficient knowledge of the observation space cannot complete these tasks. In the experiments results, we conduct experiments with tasks of increasing complexity (e.g., Figure 1), showing that the more difficult the task is, the more advantageous our approach becomes.

### F.1 Environment Details

#### F.1.1 Single Agent Scenarios

**Single Agent Maze Task:** This is a tabular-case maze task. Consider a $W$ by $H$ grid world, where landmark 1 with a high reward of $r_1$ is always placed at the higher left corner $(1, 1)$, while landmark 2 with a low reward of $r_2$ is always placed at the middle diagonal position $([W/2], [H/2])$. There are $N_o$ obstacles that are randomly placed on the map and there is a collision penalty (negative reward signal $r_3 = -r_2$) if the agent covers the obstacles. There is no movement cost. The task is completed and the reward $r_1$ (or $r_2$) is achieved only if the agent occupies one of the rewarding landmarks before or at the end of the game. Otherwise. The game terminates in $T$ steps or when the task is completed. The maximum training episodes are $N_e$.

- Simple Maze: $W = 4, H = 4$, there is only one target landmark placed in the top left corner and there are no obstacles. $T = 3$ and $N_e = 3000$.

- Medium Maze: $W = 8, H = 8$, there is only one target landmark placed in the top left corner and there are two obstacles. $T = 8$ and $N_e = 5000$.

- Hard Maze: $W = 10, H = 10$, there are two target landmarks placed in the top left corner and in the middle of the maze map, there are two obstacles, both are close to the higher rewarding states. $T = 10$ and $N_e = 5000$.

**D4RL- Hopper:** This environment is based on the work done by Erez, Tassa, and Todorov in [21]. The environment aims to increase the number of independent state and control variables as compared to the classic control environments. The hopper is a two-dimensional one-legged figure that consist of four main body parts - the torso at the top, the thigh in the middle, the leg in the bottom, and a single foot on which the entire body rests. The goal is to make hops that move in the forward (right) direction by applying torques on the three hinges connecting the four body parts. More details can be found in https://gymnasium.farama.org/environments/mujoco/hopper/.

**D4RL- Half Cheetah:** This environment is based on the work [63]. The HalfCheetah is a 2-dimensional robot consisting of 9 body parts and 8 joints connecting them (including two paws). The goal is to apply torque on the joints to make the cheetah run forward (right) as fast as possible, with a positive reward allocated based on the distance moved forward and a negative reward allocated for moving backward. The torso and head of the cheetah are fixed, and the torque can only be applied on the other 6 joints over the front and back thighs (connecting to the torso), shins (connecting to the thighs), and feet (connecting to the shins). More details can be found in https://gymnasium.farama.org/environments/mujoco/half_cheetah/.

#### F.1.2 Multi Agent Scenarios

**Predator-prey maze version:** agents must reach the goal simultaneously to complete this task while avoiding collisions between agents and the obstacles. Note that these tasks are quite **challenging**: (1) We increase the maze dimension and place the high-reward goal at the corner of the maze surrounded by high-penalty obstacles which are harder for agents to reach; (2). agents' observations do not include any information about the goal and other agents, so they need to grow DTs iteratively based on their joint action value vectors and get the information about their joint observation space to find the rewarding state.

Each time an agent covers with a target, the agent is rewarded with $+r_1$, if the agents reach the target at the same time, they will be given a group reward $+r_2$ or $+r_3$ based on the rewarding target type (higher rewarding state or lower rewarding state). The agents are also given a group reward $\cdot \sum_{n=1}^{N} \min\{d(\mathbf{p}_n, \mathbf{q}_k), k \in \{1, \ldots, K\}\}, n \in \{1, \ldots, N\}$, where the $\mathbf{p}_n$ and $\mathbf{q}_k$ are the coordinates

of agent $n$ and target $k$. The reward of the agents will be decreased for increased distance from targets.

## F.2 Empirical Performance Comparison between RGMDT and DRL

We show empirical performance comparisons between RGMDT and DRL in Figure 2 (single-agent task) and Figure 4 (multi-agent task) in the main body. We also add Figure 5 in Appendix F.8.2 to show that the return gap is bounded by the average cosine distance—as quantified by our analysis and theorems—and diminishes as the average cosine distance decreases due to the use of more leaf nodes (i.e., more action labels leading to lower clustering distance).

1. **Figure 2**: Shows RGMDT's enhanced performance with increasing leaf nodes, nearing optimal levels of RL with $|L| = 4$ or more. This supports Theorem 4.4's prediction that return gaps decrease as the average cosine distance is minimized.

2. **Figure 4**: Confirms RGMDT's performance improves with more leaf nodes in the multi-agent scenario, achieving near-optimal levels with $|L| = 4$ or more, consistent with the findings of Theorem 4.4 and supporting its application in multi-agent settings as noted in Remark 4.5.

3. **Figure 5**: Plots average cosine distance and the return gap between RGMDT and the expert DRL policies for different leaf node counts (8, 16, 32). The results, analyzed from the last 30 episodes post-convergence, justify Theorem 4.4's analysis: the return gaps are indeed bounded by $O(\sqrt{\epsilon/(\log_2(L+1)-1)}nQ_{\max})$, and the average return gap diminishes as the average cosine distance over all clusters decreases due to using more leaf nodes, which also validates the results in Remark 4.5.

## F.3 Comparing RGMDT with Simple DT Baselines

We include the Imitations DT baseline using CART, directly trained on the RL policy's actions and observations without resampling. The observations and actions are features and labels respectively. The results (Fig. 1-Fig. 4, Table. 1-Table. 2) demonstrate that RGMDT's superior performance becomes more noticeable with a limited number of leaf nodes since with fewer leaf nodes (or a more complex environment), some of the decision paths must be merged and some actions would change. RGMDT minimizes the impact of such changes on return.

## F.4 Other Interpretable RL baselines

RGMDT is the first work for multi-agent DT with performance guarantees. Since agents' decisions jointly affect state transition and reward, converting each agent's decision separately into DTs may not work and accurately reflect the intertwined decision-making process. Our work is able to solve this problem and provide guarantees. Thus, we didn't find other interpretable multi-agent baselines with performance guarantees except for MA-VIPER and its baselines which have been compared with RGMDT in current evaluations.

## F.5 Other Evaluating Environments

When evaluating RGMDT, we reviewed how other DT-based models were tested. The VIPER [3] evaluated the algorithm in Atari Pong, and cart-pole environments, which are much simpler environments than ours, they also evaluated the algorithm in Half-cheetah tasks which are included in our D4RL tasks. The MA-VIPER [48] only evaluated their DT algorithms in MPE environment [40] for multi-agent scenarios, in which we implement the same Predator-prey maze tasks using the same settings. We note that most existing papers evaluate DT-based algorithms only on classification datasets [33, 72]. In contrast, our evaluation includes D4RL environments, which is a more complex environment widely used for evaluating RL algorithms (and not just DT-based methods) [65]. Our evaluations show that the extracted DTs (both single- and multi-agent) is able to achieve similar performance comparable to DRL algorithms on complex D4RL environments. In the future, we could consider applying RGMDT to simulated autonomous driving and healthcare scenarios where insight into a machine's decision-making process is important, and human operators must be able to follow step-by-step procedures that can be provided with DT.

## F.6 Impact of Leaf Node Counts on RGMDT's Performance

Remark 4.5 shows that Theorem 4.4 holds for any arbitrary finite number of leaf nodes $L$. Furthermore, increasing the maximum number of leaf nodes $L$ reduces the average cosine distance (since more

clusters are formed) and, consequently, a reduction in the return gap due to the upper bound derived in Theorem 4.4. Evaluation results for varying numbers of leaf nodes: Specifically, in Figure. 2 (single-agent tasks), Figure. 3-Figure. 4 (multi-agent tasks), and Table. 1 (D4RL tasks) we show RGMDT's performance improves with an increasing number of leaf nodes, which is consistent with the findings of Theorem 4.4 and Remark 4.5 in both single- and multi-agent tasks.

### F.7 Comparison with Other Interpretable RL Methods

RGMDT is the first multi-agent DT model with performance guarantees. Since the agent's decisions jointly affect state transition and reward, converting each agent's decision separately into DTs may not work and accurately reflect the intertwined decision-making process. RGMDT is able to solve this problem and provide guarantees. RGMDT offers a more interpretable form of policy representation by directly mapping observations to actions. This differs significantly from:

1. Reward Decomposition: While this method breaks down the reward function into components for clarity, it lacks in providing a straightforward explanation of decision processes or translating complex data into a simple, explainable policy structure like DTs.

2. Option Discovery: It focuses on identifying sub-policies or "options" within a complex RL policy. These options can be seen as temporally extended actions, providing a higher-level understanding of the policy structure. However, it focuses on identifying such skill components, which could still be represented by a deep skill policy, e.g., in deep option discovery, and is less interpretable than a decision tree, which provides clear decision paths based on observations.

### F.8 Addtional Ablation Studies

We added a new set of experiments to assess how errors of non-Euclidean clustering and return gaps influence the algorithm's performance,

### F.8.1 Experiment 1. Ablation Study on Non-Euclidean Clustering Error Impact

We conducted experiments for Ablation Study using two other clustering metrics Euclidean and Manhattan distances to replace the Non-Euclidean Clustering metrics cosine distance used in the original RGMDT. We also introduced noise levels from 0.0 to 0.6 to simulate various environmental conditions and compared the performance of different clustering metrics across the same error levels to assess how each metric manages increasing noise.

Table. 3 shows that cosine distance exhibits high resilience, maintaining significantly higher rewards than the other metrics up to an error level of 0.6, indicating robust performance. In contrast, Euclidean distance consistently underperformed, failing to meet targets even at zero error, highlighting its inadequacy for tasks requiring non-Euclidean measures. Meanwhile, Manhattan distance performed better than Euclidean but was still inferior to RGMDT, with performance dropping as error increased, thus confirming Theorem 4.4, minimizing cosine distance effectively reduces the return gap. The ablation study confirms that non-Euclidean clustering errors significantly impact the performance, with RGMDT showing notable robustness, particularly under higher error conditions, highlighting its capacity to utilize the geometric properties of sampled Q-values for DT construction. This is because RGMDT aims to group the observations that follow the same decision path and arrive at the same leaf node (which is likely maximized with the same action, guided by action-values ) in the same cluster, therefore, non-euclidean clustering metric cosine distance is more efficient here (proven in Theorem 4.4).

### F.8.2 Experiment 2. Impact of Return Gaps Errors on Performance

The goal of RGMDT is to minimize return gaps. So we could only conduct an additional experiment demonstrating that return gaps decrease as RGMDT minimizes cosine distance. Since minimizing return gaps is our primary objective, their initial measurement isn't possible until after RGMDT training.

Directly minimizing the return gap is challenging, but Theorem 4.4 proves that it is bounded by $O(\sqrt{\epsilon/(\log_2(L+1)-1)}nQ_{\max})$, where $\epsilon$ is the average cosine-distance within each cluster. This finding motivates the RGMDT, which reduces the return gap by training a non-Euclidean clustering

network $g$ to optimize $\epsilon$ with a cosine-distance loss. Thus, RGMDT effectively minimizes the return gap error.

Fig. 5 in the rebuttal PDF shows the correlation between average cosine-distance $\epsilon$ and the return gap across clusters with 8, 16, and 32 leaf nodes. We calculated the return gap using mean episode rewards from the last 30 episodes after RGMDT and expert RL policies converged. The findings justify Theorem 4.4's analysis that the return gap decreases as $\epsilon$ reduces with more leaf nodes, validating the results in Remark 4.5.

### F.9 Interpretability of RGMDT

Note that RGMDT is the first work for multi-agent DT with performance guarantees. Since the agents' decisions jointly affect state transitions and rewards, converting each agent's decision separately into decision trees may not accurately reflect the intertwined decision-making process. Our work addresses this problem and provides theoretical guarantees.

RGMDT enhances the interpretability of DRL policies by translating them into DT structures. Small decision trees (with tree depth less than 6) are generally considered interpretable because they provide a clear, step-by-step representation of decision-making processes based on input features, as further discussed in [48].

Key aspects enhancing RGMDT's interpretability include:

1. Clear Decision Paths: DTs offer explicit paths from root to leaf nodes, each linked to a specific action, unlike the complex approximations in DRL policies.

2. Quantifiable Performance Guarantee: RGMDT quantifies the return gap between the DRL and DT policies, ensuring a reliable measure of how closely the DT mimics the DRL policy.

3. Interpretability at Different Complexity Levels: RGMDT can generate a DT with a return gap guarantee for any decision tree size, providing a trade-off between interpretability and accuracy. A smaller DT offers better interpretability at the cost of a higher return gap. Regardless, our method ensures theoretical guarantees on the return gap. According to [48], DTs with a tree depth less than 6 ($2^6 = 64$ leaf nodes) can be considered naturally interpretable. Notably, with only more than $|L| = 4$ leaf nodes, RGMDT achieves near-optimal performance of the expert DRL policy in both single-agent and multi-agent scenarios, while other baselines fail to complete the task (as supported by results in Figure 2 and Figure 4).

4. Cluster-Based Approach: RGMDT clusters observations into different decision paths in the DT, grouping similar decisions and making it easier to understand which types of observations lead to specific actions.

We have provided some DT visualizations of maze tasks in Appendix F.11, illustrating how agents make decisions based on their observations (e.g., location coordinates in this task). A more concrete example is given in Figure 7 to Figure 10, Appendix F.11: the features include agents' X and Y coordinates. For this 4-node DT, the right decision path shows that if the agent's Y coordinate is larger than $0.4652$, the decision goes to the right node. By further splitting based on the agent's X coordinates (if $X$ is larger than $0.0125$), two decision paths indicate taking actions to go 'right' ($X \leq 0.0125$) or 'up' ($X > 0.0125$).

### F.10 Additional Experiments on Interpretability

To illustrate non-euclidean clustering labels interpretation, we run RGMDT on a two-agent grid-world maze for easy visualization and add four more figures (Fig. 6) to further visualize the relationships between: 1. action and labels; 2. position and labels, since DT generation is guided by non-euclidean clustering results.

Fig. 6 shows that the non-euclidean clustering labels used in RGMDT are naturally interpretable. We explored the relationship between non-Euclidean clustering labels, agent positions, and actions. Fig. 6a and Fig. 6b show how agent positions during training correlate with clustering labels: 'blue', 'green', 'pink', and 'orange' indicate labels '0', '1', '2', and '3', respectively. Agents near the high-reward target are typically labeled '1', while those near the lower-reward target get labeled '2'. Additionally, Fig. 6c and Fig. 6d demonstrate that agents take actions 'down', 'up', 'right', and 'left' conditioned on specific labels '0', '1', '2', '3', respectively. Putting these together, when an RGMDT

Table 3: The superior performance of RGMDT when using non-Euclidean cosine-distance metrics indicates that the algorithm effectively utilizes the geometric properties of sampled Q-values for DT construction. This is supported by our analysis which links RGMDT's robustness to its ability to minimize the return gap using cosine distance. This advantage becomes particularly evident under higher error conditions, where traditional Euclidean and other non-Euclidean metrics fail. As errors increase, a performance drop across all metrics is typical, highlighting the importance of non-Euclidean clustering. RGMDT's design to cluster observations following similar decision paths into the same leaf nodes—typically maximized by the same actions as guided by the action-value $Q(s, a)$—enhances its effectiveness. This approach confirms that the cosine-distance metric is more suitable for maintaining performance in the face of clustering errors, as proven in Theorem 4.4.

| Clustering Metrics | Error Level | If Reached Target | Mean Reward |
|---|---|---|---|
| Non-Euclidean (RGMDT) | 0.0 | **Yes** | $\mathbf{11.65 \pm 0.88}$ |
| Non-Euclidean (RGMDT) | 0.1 | **Yes** | $9.25 \pm 0.65$ |
| Non-Euclidean (RGMDT) | 0.2 | **Yes** | $8.67 \pm 0.77$ |
| Non-Euclidean (RGMDT) | 0.4 | **Yes** | $6.34 \pm 0.58$ |
| Non-Euclidean (RGMDT) | 0.6 | No | $2.29 \pm 0.54$ |
| Euclidean | 0.0 | No | $3.66 \pm 1.03$ |
| Euclidean | 0.1 | No | $3.32 \pm 0.75$ |
| Euclidean | 0.2 | No | $2.82 \pm 0.82$ |
| Euclidean | 0.4 | No | $2.03 \pm 0.65$ |
| Euclidean | 0.6 | No | $-1.62 \pm 0.92$ |
| Manhattan | 0.0 | **Yes** | $5.67 \pm 0.93$ |
| Manhattan | 0.1 | No | $4.52 \pm 1.22$ |
| Manhattan | 0.2 | No | $3.84 \pm 0.84$ |
| Manhattan | 0.4 | No | $2.13 \pm 0.75$ |
| Manhattan | 0.6 | No | $0.92 \pm 1.22$ |

agent approaches the high-reward target, the position will be labeled as '1', instructing other agents to move 'up' or 'left', which effectively guides other agents to approach the high-reward target located at the upper left corner of the map, influencing strategic movements towards targets.

### F.11   Decision Tree Visualization

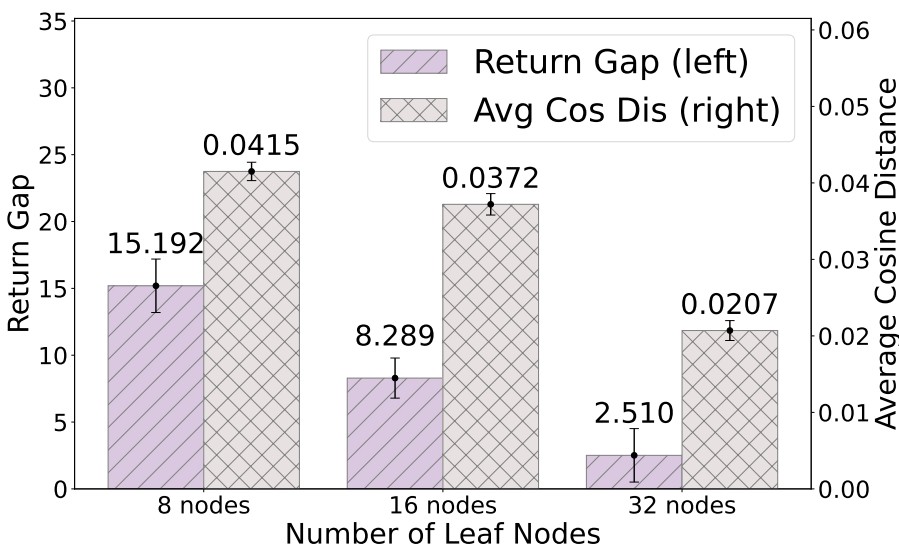

Figure 5: The return gap (left) is bounded by average cosine distance (right) and diminishes as average cosine distance (right) decreases due to the increase of the maximum number of leaf nodes.

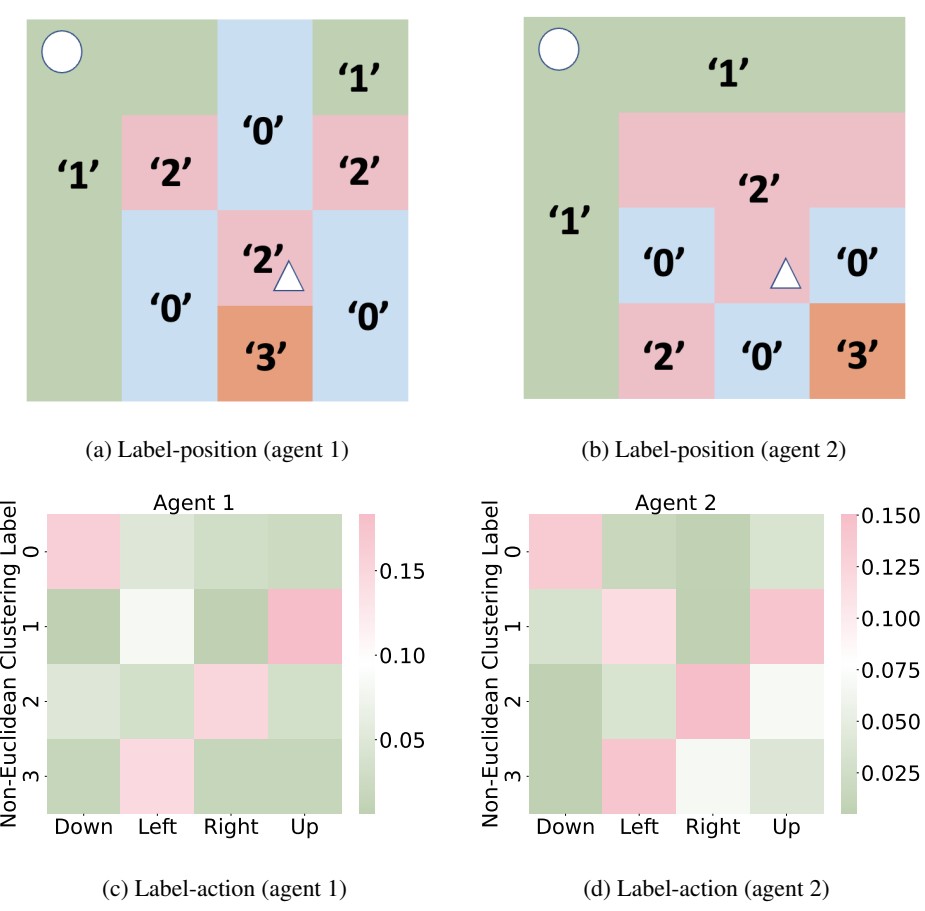

(a) Label-position (agent 1)          (b) Label-position (agent 2)

(c) Label-action (agent 1)          (d) Label-action (agent 2)

Figure 6: Interpretable Non-Euclidean Clustering Labels: (a)-(b): The positions are more likely to be labeled as '1' when closer to the higher-reward target (circle), while more likely to be labeled as '2' when closer to the lower-reward target (triangle); (c)-(d): both agents are more likely to take action 'down', 'up', 'right', 'left' conditioned on labels '0', '1', '2', and '3', respectively.

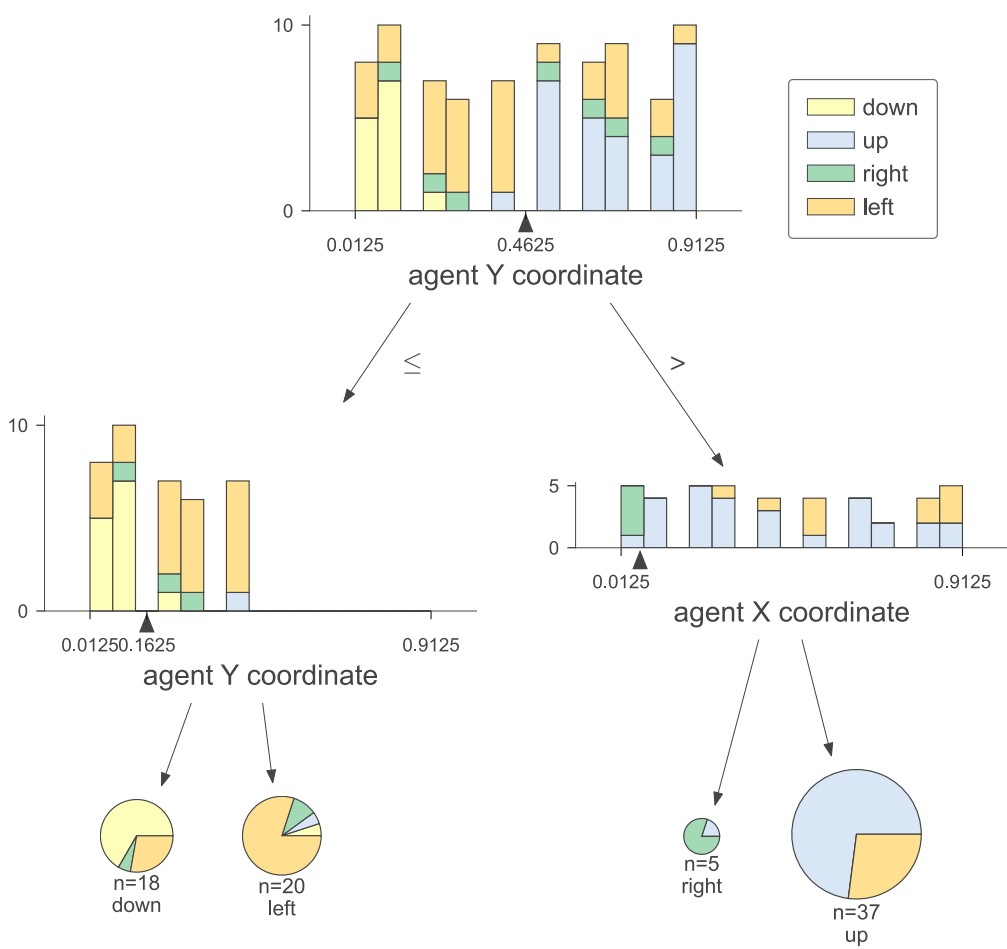

Figure 7: Visulazing the RGMDT decision path for Hard Maze with the maximum number of leaf nodes equals 4.

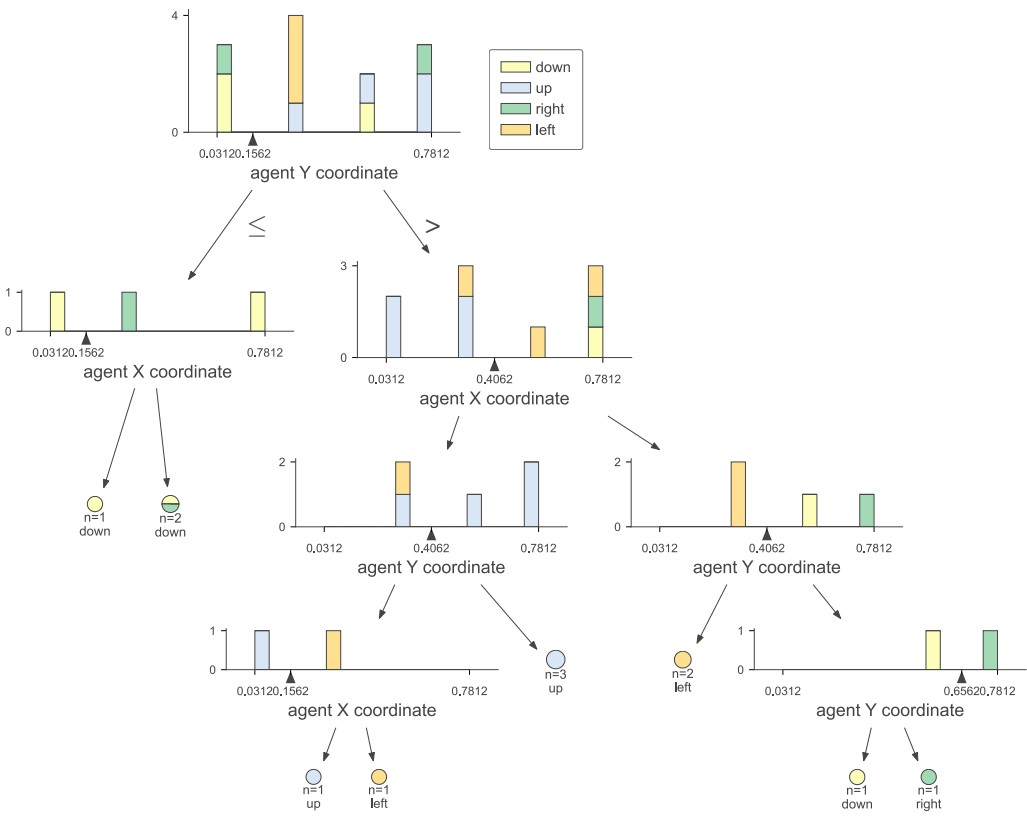

Figure 8: Visulazing the RGMDT decision path for Simple Maze with the maximum number of leaf nodes equals 8.

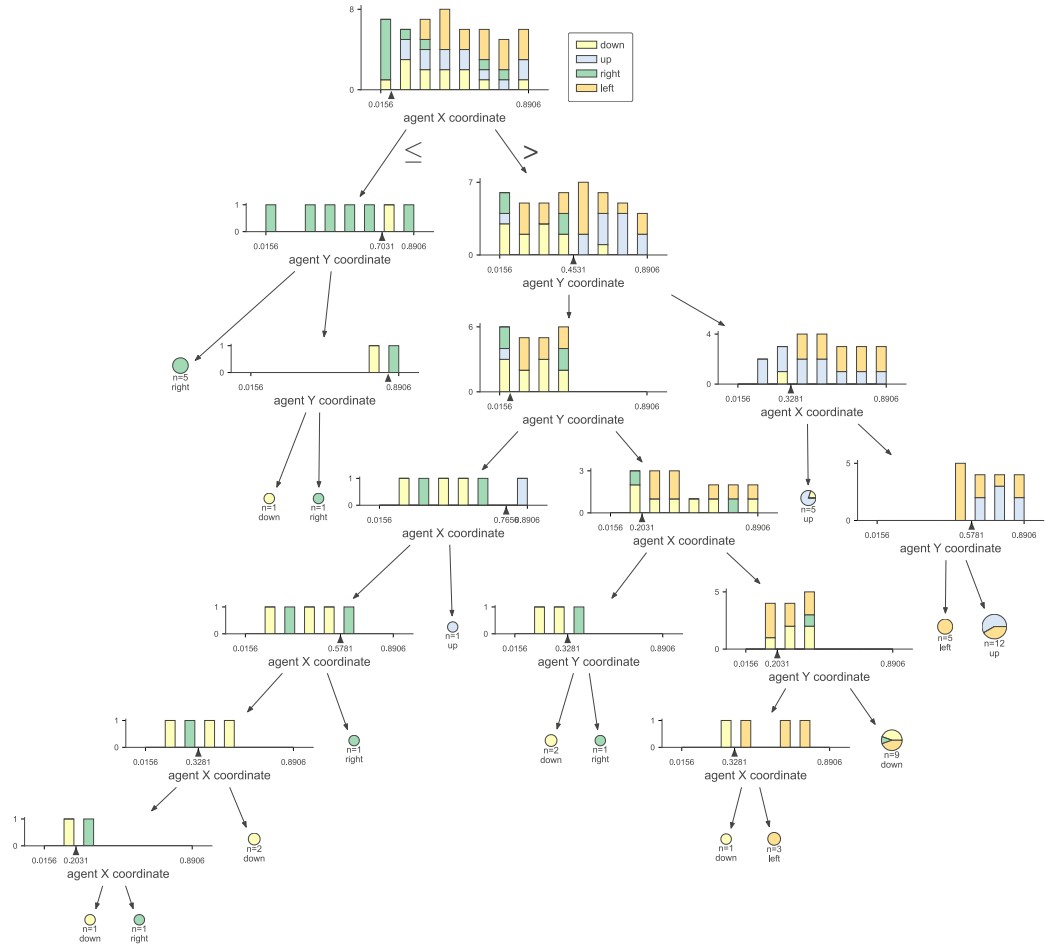

Figure 9: Visulazing the RGMDT decision path for Medium Maze with the maximum number of leaf nodes equals 16.

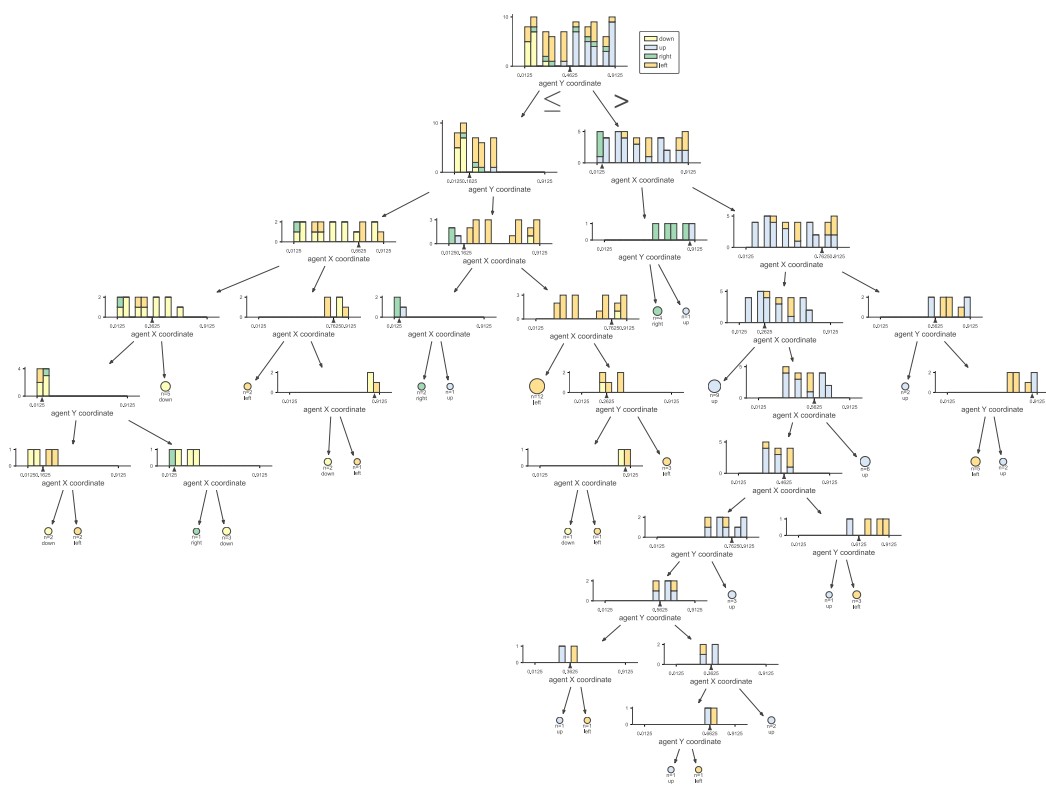

Figure 10: Visulazing the RGMDT decision path for Hard Maze with the maximum number of leaf nodes equals 32.

