# OpenReview forum: "RGMDT: Return-Gap-Minimizing Decision Tree Extraction in Non-Euclidean Metric Space"
_NeurIPS.cc/2024/Conference — NeurIPS 2024 poster_

### Official Review · Reviewer_Cxpi · 2024-07-12

**Soundness:** 3
**Presentation:** 3
**Contribution:** 3
**Rating:** 7
**Confidence:** 3

**Summary:**

This work provides an upper bound on the return gap between the DRL policy and its extracted DT policy. Based on this, it proposes the RGMDT algorithm with a simple design that can be extended to multi-agent settings using an iteratively-grow-DT procedure. The RGMDT algorithm outperforms other DT-based algorithms in the D4RL tasks.

**Strengths:**

1. This work proposes the Return-Gap-Minimization Decision Tree (RGMDT), which minimizes the return gap between itself and the DRL policy for any given size.

2. The RGMDT can be extended to a multi-agent framework using an iterative non-Euclidean clustering formulation.

3. The RGMDT achieves more promising performance than other DT-based algorithms in D4RL tasks.

**Weaknesses:**

One of the motivations of this work is that DRL policies cannot be interpreted and understood well, and that decision trees (DT) could help improve interpretability. However, I don't see how the proposed method, RGMDT, enhances interpretability. This work seems to only demonstrate the superior performance of RGMDT compared to other DT-based algorithms, without addressing how it aids in the interpretability of DRL policies.

**Questions:**

1. This work provides an upper bound on the return gap between the DRL policy and its extracted RGMDT from a theoretical perspective. However, how does the empirical performance compare?

2. My main concern is how RGMDT can interpret and understand the policy of DRL. How does RGMDT enhance the interpretability of DRL policies?

**Limitations:**

There could be more discussion to the limitations

---

> ### Author Rebuttal · Authors · 2024-08-04
>
> # Interpretability of RGMDT
> Note that RGMDT is **the first work for multi-agent DT with performance guarantees**. Since the agent's decisions jointly affect state transition and reward, converting each agent's decision separately into decision trees may not work and accurately reflect the intertwined decision-making process.  Our work can solve this problem and provide guarantees.
>
> RGMDT enhances the interpretability of DRL policies by translating them into DT structures. **Small Decision trees (tree depth less than 6) are generally considered interpretable** because they provide a clear, step-by-step representation of decision-making processes based on input features, as further discussed in [1] and will be added in our final version.
>
> **Key aspects enhancing RGMDT's interpretability include**:
> 1. **Clear Decision Paths**: DTs offer **explicit paths** from the root to leaf nodes, each **linked to a specific action**, unlike the complex approximations in DRL policies.
> 2. **Quantifiable Performance Guarantee**: RGMDT **quantifies the return gap** between the DRL and DT policies, ensuring **a reliable measure** of how closely the DT mimics the DRL policy.
> 3. **Interpretability at different complex levels**: Since RGMDT can generate a DT with a return gap guarantee for **any decision tree size**, it provides a **trade-off between complexity level and accuracy of interpretability**. A smaller DT can offer better interpretability at the cost of a higher return gap. In any case, our method ensures theoretical guarantees on the return gap. According to [1], **DTs with tree depth less than 6 ($2^6=64$ leaf nodes) can be considered naturally interpretable**. Notably, **with only more than $|L|=4$ leaf nodes**, **RGMDT achieves near-optimal performance of the expert DRL policy** in **single-agent** and **multi-agent** scenarios while **other baselines fail to complete the task** (**supported by results in Fig.2 and Fig.4**).
> 4. **Cluster-Based Approach**: The method's foundation in **clustering observations into different paths** in the DT helps **group similar decisions**, making it easier to **understand which types of observations lead to which actions**.
>
> We have provided some **DT visualization** of maze tasks in **Appendix. E.2**. It shows **how agents make decisions based on their observations** (location's coordinates in this task).
> Give a more **concrete example**, like in **Fig.5, Appendix E.2**, the features include agents' X and Y coordinates, For this 4-node DT, the right decision path shows that if the agent's Y coordinate is larger than $0.4652$, the decision goes to the right node. By further splitting it according to the agent's X coordinates (if $X$ is larger than $0.0125$), two decision paths are indicating taking actions going 'right' ($X \leq 0.0125$) or going 'up' ($X > 0.0125$).
>
> [1] Stephanie Milani, et al. Maviper: Learning decision tree policies for interpretable multi-agent reinforcement learning, 2022.
>
> # Additional Experiments on Interpretability
> To illustrate **non-euclidean clustering labels interpretation**, we run RGMDT on a two-agent grid-world maze for easy visualization and **add four more figures** (see **Fig.2 in the rebuttal PDF file**) to further visualize the relationships between: **1.action and labels**; **2.position and labels**, since DT generation is guided by non-euclidean clustering results.
>
> **Fig.2 in rebuttal PDF** shows that the non-euclidean clustering labels used in RGMDT are naturally **interpretable**. We explored the relationship between non-Euclidean clustering labels, agent positions, and actions. **Fig.2(a)** and **Fig.2(b)** show how agent positions during training correlate with clustering labels: 'blue', 'green', 'pink', and 'orange' indicate labels '0', '1', '2', and '3', respectively. Agents near the high-reward target are typically labeled '1', while those near the lower-reward target get labeled '2'. Additionally, **Fig.2(c)** and **Fig.2(d)** demonstrate that agents take actions 'down', 'up', 'right', 'left' conditioned on specific labels '0', '1', '2', '3', respectively. Putting these together, when an RGMDT agent approaches the high-reward target, the position will be labeled as '1', instructing other agents to move 'up' or 'left', which effectively guides other agents to approach the high-reward target located at the upper left corner of the map, influencing strategic movements towards targets
>
> # Empirical Performance Comparison between RGMDT and DRL
> We show empirical performance comparisons between RGMDT and DRL in **Fig.2 (single-agent task)** and **Fig.4 (multi-agent task)** in the main body.
> We also add **Fig.1 in the rebuttal PDF** to show that **the return gap is bounded by average cosine distance** - as quantified by our analysis and theorems - and diminishes as average cosine distance decreases due to the use of more leaf nodes (i.e., more action labels leading to lower clustering distance).
> 1. **Fig.2**: Shows RGMDT's enhanced performance with increasing leaf nodes, nearing optimal levels of RL with $|L|=4$ or more. This supports Thm.4.4's prediction that return gaps decrease as the average cosine distance is minimized.
> 2. **Fig.4**: Confirms RGMDT's performance improves with more leaf nodes in the multi-agent scenario, achieving near-optimal levels with $|L|=4$ or more, consistent with the findings of Thm.4.4 and supporting its application in multi-agent settings as noted in Remark.4.5.
> 3. **Fig.1 in rebuttal PDF**: Plots average cosine distance and **the return gap between RGMDT and the expert DRL policies** for different leaf node counts (8, 16, 32). The results, analyzed from the last 30 episodes post-convergence, justify Theorem 4.4's analysis: the return gaps are indeed bounded by $O(\sqrt{\epsilon/(\log_{2}(L+1) - 1)}nQ_{\rm max})$, and the average return gap diminishes as the average cosine-distance over all clusters decreases due to using more leaf nodes, which also validates the results in Remark 4.5.

---

> > ### Comment · Reviewer_Cxpi · 2024-08-09
> > **Response to Author's Rebuttal**
> >
> > Thank you for your response to my comments. I have read your rebuttal and will raise my score.

---

> > > ### Author Response · Authors · 2024-08-09
> > >
> > > Dear reviewer, thank you so much for your response, we really appreciate it!

---

### Official Review · Reviewer_uDjD · 2024-07-15

**Soundness:** 3
**Presentation:** 3
**Contribution:** 3
**Rating:** 6
**Confidence:** 3

**Summary:**

The authors proposes a method called Return-Gap-Minimization Decision Tree (RGMDT) to extracting interpretable decision tree policies from learned parametric RL policies. The authors first propose a method to quantify the return gap between an oracle RL policy and its extracted decision tree policy, which provides a guarantee for the performance of the DT policy. The RGMDT is built upon the idea to minimizing this gap, which works by recasting the DT extraction problem as an iterative non-euclidean clustering problem. In this clustering problem, the goal is to clustering different observations into different decision tree paths, where the leaf nodes are corresponding to the action.  Additionally, the authors also extend the algorithm to the multi-agent setting and provide theoretical analysis for the performance guarantee as well.  Empirically, RGMDT performs much better than DT but being more interpretable than RL polices.

**Strengths:**

The proposed algorithm is theoretically sound and the analysis is well-executed. The extension to the multi-agent is great, though I would say it has very limited applicability given that multi-agent RL seem still not work well in practice. The experiments support the claim and demonstrate the effectiveness of the proposed method.

**Weaknesses:**

- 1. Although the authors have compared RGMDT to several DT extraction baselines, it doesn't compare with such a baseline: train a simple decision tree on the RL policy's action and observations. This should be doable and is an important baseline to compare against.


- 2. Some ablation studies are missing. For example, it would be great if the authors could ablate how the error of non-Euclidean clustering and the return gap impact the final performance. How sensitive is the algorithm to these errors?


**Questions**:

- a. How does the performance of RGMDT change with the size of the DT (like, the maximum number of leaf nodes)? The paper shows results for different maze sizes, but it would be nice to see the trend more systematically.

- b. How does RGMDT compare to other interpretable RL approaches, such as reward decomposition or option discovery?

**Questions:**

See weakness.

**Limitations:**

See weakness.

---

> ### Author Rebuttal · Authors · 2024-08-04
>
> # Query on Simple DT Baselines
> We include the requested **Imitations DT** baseline using **CART**, directly trained on the **RL policy's actions and observations** (lines 265-271) as described, without resampling. The observations and actions are **features** and **labels** respectively. The **results (Fig.1-4, Table.1-2)** demonstrate that RGMDT's superior performance becomes more noticeable with a limited number of leaf nodes since with fewer leaf nodes (or a more complex environment), some of the decision paths must be merged and some actions would change. RGMDT minimizes the impact of such changes on return.
> # Query on Ablation Studies
> We added **a new set of experiments** to assess how errors of non-Euclidean clustering and return gaps influence the algorithm’s performance, detailed in **Table.1** and **Fig.1** in the **rebuttal PDF file**.
> ## Experiment 1: Ablation Study on Non-Euclidean Clustering Error Impact
> We conducted experiments for **Ablation Study** using two other clustering metrics **Euclidean and Manhattan distances** to replace **the Non-Euclidean Clustering metrics cosine distance** used in the original RGMDT. We also introduced **noise levels from 0.0 to 0.6** to simulate various environmental conditions and compared the performance of different clustering metrics across the same error levels to assess how each metric manages increasing noise.
>
> **Table.1 in the rebuttal PDF file** shows that **cosine distance** exhibits high resilience, maintaining **significantly higher rewards** than the other metrics up to an error level of 0.6, indicating **robust** performance. In contrast, **Euclidean distance** consistently underperformed, failing to meet targets even at zero error, highlighting its inadequacy for tasks requiring non-Euclidean measures. Meanwhile, **Manhattan distance** performed better than Euclidean but was still inferior to RGMDT, with performance dropping as error increased, thus **confirming Thm. 4.4** that **minimizing cosine distance effectively reduces the return gap**. The ablation study confirms that **non-Euclidean clustering errors** significantly impact the performance, with **RGMDT showing notable robustness**, particularly **under higher error conditions**, highlighting its capacity to utilize the geometric properties of sampled Q-values for DT construction. This is because RGMDT aims to group the observations that follow the same decision path and arrive at the same leaf node (which are likely maximized with the same action, guided by action-values $Q(s,a)$) in the same cluster, therefore, **non-euclidean clustering metric cosine-distance** is more **efficient** here (**proven in Thm. 4.4**).
> ## Experiment 2: Impact of Return Gaps Errors on Performance
> **The goal of RGMDT is to minimize return gaps**. So we could only conduct **an additional experiment** demonstrating that **return gaps decrease as RGMDT minimizes cosine-distance**. Since minimizing return gaps is our **primary objective**, their initial measurement isn't possible until after RGMDT training.
>
> Directly minimizing the **return gap** is challenging, but **Thm. 4.4** proves that it is bounded by $O(\sqrt{\epsilon/(\log_{2}(L+1) - 1)}Q_{\rm max})$, where $\epsilon$ is the average **cosine-distance** within each cluster. This finding motivates the RGMDT, which **reduces the return gap** by training a non-Euclidean clustering network $g$ to optimize $\epsilon$ **with a cosine-distance loss**. Thus, RGMDT effectively **minimizes the return gap error**.
>
> **Fig.1 in the rebuttal PDF** shows the correlation between **average cosine-distance $\epsilon$** and the return gap across clusters with 8, 16, and 32 leaf nodes. We calculated the return gap using mean episode rewards from the last 30 episodes after RGMDT and expert RL policies converged. The findings **justify Thm. 4.4's analysis** that the return gap decreases as $\epsilon$ reduces with more leaf nodes, **validating the results in Remark 4.5**
> # Impact of Leaf Node Counts on RGMDT's Performance
> **Remark 4.5 (line 200)** shows that **Thm.4.4** holds for **any arbitrary finite number of leaf nodes $L$**. Furthermore, increasing the maximum number of leaf nodes $L$ reduces the average cosine distance (since more clusters are formed) and, consequently, a reduction in the return gap due to the upper bound derived in Thm.4.4.
> **Evaluation results for varying numbers of leaf nodes:** Specifically, in **Fig.2 (single-agent tasks), Fig.3-4 (multi-agent tasks), and Table 1 (D4RL tasks)** we show **RGMDT's performance improves with an increasing number of leaf nodes**, which is consistent with the findings of Thm.4.4 and Remark 4.5 **in both single- and multi-agent tasks**.
> # Comparison with Other Interpretable RL Methods
> RGMDT is the **first multi-agent DT model with performance guarantees**. Since the agent's decisions jointly affect state transition and reward, converting each agent's decision separately into DTs may not work and accurately reflect the intertwined decision-making process. RGMDT is able to solve this problem and provide guarantees.
> RGMDT offers a more interpretable form of policy representation by directly mapping observations to actions. This differs significantly from:
> 1. **Reward Decomposition**: While this method breaks down the reward function into components for clarity, **it lacks in providing a straightforward explanation of decision processes or translating complex data into a simple, explainable policy structure like DTs**.
> 2. **Option Discovery**: It focuses on identifying sub-policies or "options" within a complex RL policy. These options can be seen as temporally extended actions, providing a higher-level understanding of the policy structure. However, it focuses on identifying such skill components, **which could still be represented by a deep skill policy, e.g., in deep option discovery**, and is less interpretable than a decision tree, which provides clear decision paths based on observations.

---

> > ### Comment · Reviewer_uDjD · 2024-08-09
> >
> > Thanks for the authors' responses the the additional experiments. I don't have additional concerns, and thus I decided to keep the rating.

---

> > > ### Author Response · Authors · 2024-08-09
> > >
> > > Dear reviewer, thank you so much for your response!

---

### Official Review · Reviewer_p7eT · 2024-08-01

**Soundness:** 3
**Presentation:** 2
**Contribution:** 3
**Rating:** 5
**Confidence:** 2

**Summary:**

This paper considers extracting decision tree (DT) based policies from DRL policies for the purpose of interpretability. The authors present an upper bound on the return gap of the oracle policy and the DT policy, which helps formulate the DT extraction problem into a non-euclidean clustering problem. The authors then propose a multi-agent variant with an iteratively-grow-DT procedure, and propose a practical algorithm (RGMDT) which outperform heuristic baselines in the maze the D4RL benchmark.

**Strengths:**

- Methodology: Using clustering to reformulate DT extraction is a fresh perspective.
- Theoretical guarantee: The paper provides a formal guarantee on the proposed algorithm.
- Flexibility: The proposed algorithm is applicable in both single-agent and multi-agent settings.
- Experiments: The paper presents concrete performance improvement over selected baselines.

**Weaknesses:**

- Presentation: I suggest the authors to enhance the clarity of the writing. More intuitive illustrations can greatly enhance the paper's impact. Also, it would be helpful to move some variants of figures in Appendix E2 in the main body of the paper to better illustrate the idea.
- Related works: I suggest the authors to add a section, either in the main body or the appendix, to sufficiently discuss existing works including interpretable RL. For example, some relevant early works, e.g., [Frosst & Hinton (2017)](https://arxiv.org/pdf/1711.09784) and [Ding et al. (2021)](https://arxiv.org/pdf/2011.07553), are missing in the paper.

References:
[1] Frosst, Nicholas, and Geoffrey Hinton. "Distilling a neural network into a soft decision tree." arXiv preprint arXiv:1711.09784 (2017).
[2] Ding, Zihan, Pablo Hernandez-Leal, Gavin Weiguang Ding, Changjian Li, and Ruitong Huang.  "Cdt: Cascading decision trees for explainable reinforcement learning." arXiv preprint arXiv:2011.07553 (2020).
[3] Milani, Stephanie, Nicholay Topin, Manuela Veloso, and Fei Fang. "Explainable reinforcement learning: A survey and comparative review." ACM Computing Surveys 56, no. 7 (2024): 1-36.

**Questions:**

- Are there any other competitive (DT-based or non DT-based) baselines for interpretable RL not covered in the current experiments of the paper? Are there any other more realistic environments beyond maze and D4RL that can be used?
- Can the authors comment on the computational complexity (in terms of time and space) of the proposed algorithm? How scalable is the proposed algorithm to real-world applications, e.g., robotics?

**Limitations:**

N/A.

---

> ### Author Rebuttal · Authors · 2024-08-04
>
> # Interpretability Presentation
> We will **move DT visualization to the main body** to better illustrate RGMDT's interpretability. To illustrate non-euclidean clustering labels interpretation, we run RGMDT on a 2-agent grid-world maze for easy visualization and **add four additional figures** (**Fig.2 in the rebuttal PDF**) to visualize the relationships between: **1.action and labels**; **2. position and labels**, since RGMDT's generation is guided by non-euclidean clustering results.
> **Fig.2 in rebuttal PDF** shows that the non-euclidean clustering labels used in RGMDT are naturally **interpretable**. We explored the relationship between position- and action-labels. **Fig.2(a)** and **Fig.2(b)** show how agent positions during training correlate with clustering labels: 'blue', 'green', 'pink', and 'orange' indicate labels '0', '1', '2', and '3', respectively. Agents near the high-reward target are typically labeled '1', while those near the lower-reward target are labeled '2'. Additionally, **Fig.2(c)** and **Fig.2(d)** shows that actions such as 'down', 'up', 'right', 'left' align with specific labels, influencing strategic movements towards objectives. For example, an agent labeled '1' near a high-reward target suggests movements 'up' or 'left', guiding others towards the high-reward target in the upper left corner of the map.
> # Interpretable RL Related Work
> We **initially included a discussion** on interpretable RL but removed it **due to page limit**. Due to 6000-character limit per response, we can't include it here. **Please see the global rebuttal for the related work section ([1-5] for interpretable RL, [6-14] for tree-based models, [6],[12],[14] for 3 mentioned references).**
> # Other Interpretable RL baselines
> RGMDT is the first work for **multi-agent DT with performance guarantees**. Since agents' decisions jointly affect state transition and reward, converting each agent's decision separately into DTs may not work and accurately reflect the intertwined decision-making process.  Our work is able to solve this problem and **provide guarantees**. Thus, we **didn't find other interpretable multi-agent baselines with performance guarantees** except for MA-VIPER and its baselines which **have been compared with RGMDT in current evaluations**.
> # Other Evaluating Environments
> When evaluating RGMDT, we reviewed **how other DT-based models were tested**. The VIPER _O.Bastani et al. Verifiable reinforcement learning via policy extraction.NeurIPS,2018_ evaluated the algorithm in **Atari Pong, cart-pole** environments, which are **much simpler environments than ours**, they also evaluated the algorithm in Half-cheetah tasks which **is included in our D4RL tasks**. The MA-VIPER _S. Milani et al. Maviper: Decision tree policies for multi-agent RL,2022_ only evaluated their DT algorithms in **MPE environment** _R. Lowe et al. Multi-agent actor-critic for mixed environments.NeurIPS,2017_ **for multi-agent scenarios**, in which we implement the **same Predator-prey** maze tasks **use the same settings**, see experimental details in **Appendix F.1.2**. We note that most existing papers evaluate DT-based algorithms **only on classification datasets** _X. Hu et al. Optimal sparse decision trees. NeurIPS,2019; H. Zhu et al. Learning optimal multivariate decision trees. NeurIPS,2020_. In contrast, our evaluation includes **D4RL environments**, which is **a more complex environment widely used for evaluating RL** algorithms (and not just DT-based methods) _H. Xu et al. A policy-guided imitation approach for offline RL.NeurIPS,2022_. Our evaluations show that the extracted DTs (both single- and multi-agent) is able to achieve similar performance comparable to DRL algorithms on complex D4RL environments. **In the future**, we could consider applying RGMDT to simulated **autonomous driving** and **healthcare** scenarios where the insight into a machine’s decision-making process is important and human operators must be able to **follow step-by-step procedures that can be provided with DT**.
> # Computational Complexity
> **Since we grow RGMDT with a small number of leaf nodes $L$, it's time- and space-efficient compared to other large DTs and DNNs**. **(1).Time Complexity**: It is determined by the Non-Euclidean clustering and DT construction steps, estimated as $O(T \cdot n^2 \cdot \log L)$, where $T$ represents the number of iterations of clustering for convergence, $n$ is the number of Q-value samples, and $L$ is the maximum number of leaf nodes. This reflects the intensive computation required for non-Euclidean distance calculations and iterative tree growth. **(2).Space Complexity**: It's $O(K(n \cdot d + L))$. This accounts for the storage of $n$ Q-value samples each with $d$ dimensions and the DT structures with $L$ leaf nodes per tree across $K$ agents.
> # Real-World Applications
> The **superior performance** of RGMDT with a small number of leaf nodes **enhances its compactness and simplifies its implementation in practical scenarios**. Compared to DNNs, its **simpler DT structure** requires **less computational and memory resources** during inference, making it well-suited for **resource-limited environments** in real-world applications like **robotics**, **network security**, and **5G network slicing resource management**. **For example**, DTs have been implemented in memristor devices to support real-time intrusion detection in scenarios requiring low latency and high speed (_Chen, J., et al. "Ride: Real-time intrusion detection via explainable machine learning implemented in a memristor hardware architecture." 2023 IEEE DSC_). RGMDT's **interpretable structure** makes it more suitable for memristor-based hardware implementations in resource-constrained environments for **network intrusion detection achieves detection speeds of microseconds, together with significant area reduction and energy efficiency**, with **performance guarantee** that previous DTs fail to provide.

---

> ### Author Response · Authors · 2024-08-12
>
> Dear Reviewer p7eT,
>
> Thank you for reviewing our paper and giving us valuable suggestions! We believe that we have addressed all the questions you asked in the review, please let us know if you have any other concerns or questions regarding our paper, we are more than happy to answer them for you. Again, thank you so much for your time and effort in reviewing our work!
>
> Warm regards :)

---

### Author Rebuttal · Authors · 2024-08-04

# Three more experiments are added in PDF
# Response to Reviewer p7eT's Query on the Interpretable RL Related Work
We have had a section discussing interpretable RL in the related work section which includes all the three mentioned references (**[6],[12],[14]**), but we deleted it due to the page limit. We will add back the following paragraph **discussing Interpretable RL in the final version**.

**Effort on Interpretability for Understanding Decisions**: To enhance interpretability in decision-making models, one strategy involves crafting interpretable reward functions within inverse reinforcement learning (IRL), as suggested by **[1]**. This approach offers insights into the underlying objectives guiding the agents' decisions. Agent behavior has been conceptualized as showing preferences for certain counterfactual outcomes **[2]**, or as valuing information differently when under time constraints **[3]**. However, extracting policies through black-box reinforcement learning (RL) algorithms often conceals the influence of observations on the selection of actions. An alternative is to directly define the agent's policy function with an interpretable framework. Reinforcement learning policies have thus been articulated using a high-level programming language **[4]**, or by framing explanations around desired outcomes **[5]**, facilitating a more transparent understanding of decision-making processes.

**Interpretable RL via Tree-based models**: To interpret an RL agent, Frosst et al **[6]** explain the decisions made by DRL policies by using a trained neural net to create soft decision trees. Coppens et al. **[7]** propose distilling the RL policy into a differentiable DT by imitating a pre-trained policy. Similarly, Liu et al. **[8]** apply an imitation learning framework to the Q-value function of the RL agent. They also introduce Linear Model U-trees (LMUTs), which incorporate linear models in the leaf nodes. Silva et al. **[9]** suggest using differentiable DTs directly as function approximators for either the Q function or the policy in RL. Their approach includes a discretization process and a rule list tree structure to simplify the trees and enhance interpretability. Additionally, Bastani et al. **[10]** propose the VIPER method, which distills policies as neural networks into a DT policy with theoretically verifiable capabilities that follow the Dataset Aggregation (DAGGER) method proposed in **[11]**, specifically for imitation learning settings and nonparametric DTs. Ding et al. **[12]** try to solve the instability problems when using imitation learning with tree-based model generation and apply representation learning on the decision paths to improve the decision tree-based explainable RL results, which could achieve better performance than soft DTs. Milani et al. extend VIPER methods into multi-agent scenarios **[13]** in both centralized and decentralized ways, they also summarize a paper about the most recent works in the fields of explainable AI **[14]**, which confirms the statements that small DTs are considered naturally interpretable.

# References

[1] Chan, Alex J., and Mihaela van der Schaar. "Scalable Bayesian inverse reinforcement learning." arXiv preprint arXiv:2102.06483 (2021).

[2]Bica, Ioana, et al. "Learning" what-if" explanations for sequential decision-making." arXiv preprint arXiv:2007.13531 (2020).

[3] Jarrett, Daniel, and Mihaela Van Der Schaar. "Inverse active sensing: Modeling and understanding timely decision-making." arXiv preprint arXiv:2006.14141 (2020).

[4] Verma, Abhinav, et al. "Programmatically interpretable reinforcement learning." International Conference on Machine Learning. PMLR, 2018.

[5] Yau, Herman, Chris Russell, and Simon Hadfield. "What did you think would happen? explaining agent behavior through intended outcomes." Advances in Neural Information Processing Systems 33 (2020): 18375-18386.

[6]Nicholas Frosst and Geoffrey Hinton. Distilling a neural network into a soft decision tree. arXiv preprint arXiv:1711.09784, 2017.

[7]Youri Coppens, Kyriakos Efthymiadis, Tom Lenaerts, Ann Nowé, Tim Miller, Rosina Weber, and Daniele Magazzeni. Distilling deep reinforcement learning policies in soft decision trees. In Proceedings of the IJCAI 2019 workshop on explainable artificial intelligence, pages 1–6, 2019.

[8]Guiliang Liu, Oliver Schulte, Wang Zhu, and Qingcan Li. Toward interpretable deep reinforcement learning with linear model u-trees. In Machine Learning and Knowledge Discovery in Databases: European Conference, ECML PKDD 2018, Dublin, Ireland, September 10–14, 2018, Proceedings, Part II 18, pages 414–429. Springer, 2019.

[9]Andrew Silva, Taylor Killian, Ivan Dario Jimenez Rodriguez, Sung-Hyun Son, and Matthew Gombolay. Optimization methods for interpretable differentiable decision trees in reinforcement learning. arXiv preprint arXiv:1903.09338, 2019.

[10]Osbert Bastani, Yewen Pu, and Armando Solar-Lezama. Verifiable reinforcement learning via policy extraction. Advances in neural information processing systems, 31, 2018.

[11]Stéphane Ross, Geoffrey Gordon, and Drew Bagnell. A reduction of imitation learning and structured prediction to no-regret online learning. In Proceedings of the Fourteenth International Conference on Artificial Intelligence and Statistics, pages 627–635. JMLR Workshop and Conference Proceedings, 2011.

[12]Zihan Ding, Pablo Hernandez-Leal, Gavin Weiguang Ding, Changjian Li, and Ruitong Huang. Cdt: Cascading decision trees for explainable reinforcement learning. arXiv preprint arXiv:2011.07553, 2020.

[13]Stephanie Milani, Zhicheng Zhang, Nicholay Topin, Zheyuan Ryan Shi, Charles Kamhoua, Evangelos E. Papalexakis, and Fei Fang. Maviper: Learning decision tree policies for interpretable multi-agent reinforcement learning, 2022.

[14]Stephanie Milani, Nicholay Topin, Manuela Veloso, and Fei Fang. Explainable reinforcement learning: A survey and comparative review. ACM Computing Surveys, 56(7):1–36, 2024.

---

### Decision · Program_Chairs · 2024-09-25

**Decision:**

Accept (poster)

**Comment:**

The paper introduces the Return-Gap-Minimization Decision Tree (RGMDT) algorithm, which extracts interpretable decision tree policies from deep reinforcement learning models, with a focus on multi-agent systems. The key innovation is the formulation of DT extraction as a non-Euclidean clustering problem, providing a formal guarantee on the return gap between the oracle policy and the extracted DT policy. This approach shows strong empirical performance, particularly in complex environments, and extends to multi-agent settings.

The paper is well-written and provides solid empirical evidence to support its claims. The theoretical foundation is robust, and the use of non-Euclidean clustering is a noteworthy contribution. However, reviewers highlighted some limitations, such as the need for clearer presentation and a more explicit focus on interpretability in practice. The initial absence of a comparison with simpler DT baselines was also noted but was addressed in the authors' rebuttal.

Overall, the approach is innovative and effective, but further refinement in presentation and interpretability demonstration could enhance the paper's impact.